# Evaluating mesoscale model predictions of diurnal speedup events in the Altamont Pass Wind Resource Area of California

Robert S. Arthur[1], Alex Rybchuk[2], Timothy W. Juliano[3], Gabriel Rios[4], Sonia Wharton[1], Julie K. Lundquist[5,2], and Jerome D. Fast[6]

[1]Lawrence Livermore National Laboratory, Livermore, CA, USA
[2]National Renewable Energy Laboratory, Golden, CO, USA
[3]National Center for Atmospheric Research, Boulder, CO, USA
[4]Program in Atmospheric and Oceanic Sciences, Princeton University, Princeton, NJ, USA
[5]Department of Mechanical Engineering, Johns Hopkins University, Baltimore, MD, USA
[6]Pacific Northwest National Laboratory, Richland, WA, USA

**Correspondence:** Robert S. Arthur (arthur7@llnl.gov)

**Abstract.** Mesoscale model predictions of wind, turbulence, and wind energy capacity factors are evaluated in the Altamont Pass Wind Resource Area of California (APWRA), where the diurnal regional seabreeze and associated terrain-driven speedup flows drive wind energy production during the summer months. Results from the Weather Research and Forecasting model version 4.4 using a novel three-dimensional planetary boundary layer (3D PBL) scheme, which treats both vertical and horizontal turbulent mixing, are compared to those using a well-established one-dimensional (1D) scheme that treats only vertical turbulent mixing. Each configuration is evaluated over a nearly 3-month-long period during the Hill Flows Study, and due to the recurring nature of the observed speedup flows, diurnal composite averaging is used to capture robust trends in model performance. Both model configurations showed similar overall skill. The general timing and direction of the speedup flows is captured, but their magnitude is overestimated within a typical wind turbine rotor layer. Both also fail to capture a persistent observed near-surface jet-like flow, likely due to limited grid resolution that is typical of mesoscale models. However, the 3D PBL configuration shows several minor improvements over the 1D PBL configuration, including improved wind speed and turbulence kinetic energy profiles during the accelerating phase of the speedup events, as well as reduced positive wind speed bias at surface stations across the APWRA region. Using a mesoscale wind farm parameterization, modeled capacity factors are also compared to monthly data reported to the U.S. Energy Information Administration (EIA) during the study period. Although the monthly trend in the data is captured, both model configurations overestimate capacity factors by roughly 7–11%. Through model evaluation, this study provides confidence in the 3D PBL scheme for wind energy applications in complex terrain and provides guidance for future testing.

## 1 Introduction

Accurate mesoscale simulations of winds in the atmospheric boundary layer are essential for wind energy resource assessment and forecasting of wind power production. However, while wind turbines are often sited in regions of complex terrain to take advantage of local wind accelerations, mesoscale models are likely to experience larger errors in these regions (Jiménez and

Dudhia, 2013; Olson et al., 2019; Chow et al., 2019; Radünz et al., 2021). Errors may result from a variety of interrelated effects, including under-resolved terrain, model numerics, and the treatment of atmospheric turbulence and its interplay with atmospheric stability and diurnal cycles.

First and foremost, complex terrain is usually under-resolved in mesoscale models, a subset of numerical weather prediction (NWP) models. Historically, NWP models were run with horizontal grid spacing on the order of 10-100 km. However, with ongoing advances in computing power, operational NWP models may now be run at higher resolution. For example, the High-Resolution Rapid Refresh model (HRRR; Benjamin et al., 2016; Dowell et al., 2022), maintained by the National Oceanographic and Atmospheric Administration (NOAA), covers the continental United States with 3 km horizontal grid spac-

ing. Recently, NWP models have been tested with 1 km or sub-kilometer grids (e.g., Olson et al., 2019), but their ability to capture local terrain-driven flow variability at the grid scale or smaller is inherently limited.

Complex-terrain errors can also result from model numerics. NWP models generally use a terrain-following coordinate system (e.g., Gal-Chen and Somerville, 1975) because it provides a straightforward implementation of surface boundary conditions. However in regions with steep terrain, the grid becomes skewed, leading to model errors that often manifest as numerical

diffusion (see, e.g., Arthur et al., 2021). A variety of approaches have been taken in the literature to address these grid-related errors, including hybrid vertical coordinate systems, improved finite difference stencils, and immersed boundary methods (see discussion in Arthur et al., 2022), but these are not a focus of the present study.

All atmospheric models require a parameterization for the effects of subgrid-scale (SGS) turbulence, and this study focuses on the treatment of atmospheric turbulence as an important source of model variability. In a mesoscale model, vertical turbulent

mixing is typically parameterized using a one-dimensional (1D) planetary boundary layer (PBL) scheme. Horizontal turbulent mixing is assumed to be small and is therefore neglected in the governing equations. This assumption is valid in coarse-grid simulations, but may be violated for higher-resolution simulations (Honnert and Masson, 2014; Mazzaro et al., 2017; Muñoz-Esparza et al., 2017; Doubrawa and Muñoz-Esparza, 2020), especially in regions with complex terrain or other sources of horizontal heterogeneity.

To address this issue, Kosović et al. (2020) and Juliano et al. (2022) implemented a three-dimensional (3D) PBL scheme within the widely used Weather Research and Forecasting model (WRF; Skamarock et al., 2019). The scheme is intended for use within the so-called turbulence gray zone (Wyngaard, 2004), within which neither traditional 1D PBL schemes nor large-eddy simulation (LES) schemes are necessarily appropriate (see further discussion in Chow et al., 2019). Gray-zone resolution is a function of atmospheric stability, with PBL depth being a proxy (e.g., Rai et al., 2019), but is typically considered to span

horizontal grid spacing of 100 m to 1 km.

The 3D PBL scheme parameterizes both vertical and horizontal turbulence shear stresses and turbulent fluxes, as well as their divergences, using the framework of Mellor and Yamada (1974, 1982), which is based on a prognostic equation for the SGS turbulence kinetic energy (TKE). In this way, the scheme is similar to the 1D Mellor-Yamada-Nakanishi-Niino (MYNN) level 2.5 model (Nakanishi and Niino, 2006) available in WRF, but with full 3D treatment of turbulent mixing. It should be noted

that with MYNN or other 1D PBL schemes, a two-dimensional (2D) form of the Smagorinsky model (Smagorinsky, 1963) is

often used to add additional horizontal diffusion and can thus be considered a form of smoothing to improve numerical stability (e.g., Smagorinsky, 1993).

In an effort to further develop the WRF 3D PBL scheme for wind energy applications, Rybchuk et al. (2022) coupled it to the mesoscale wind farm parameterization of Fitch et al. (2012). Hereafter denoted WFP, the Fitch et al. (2012) parameterization accounts for the presence of wind turbines by adding drag and TKE to the flow within the turbine rotor region. These effects are aggregated over each horizontal grid cell based on the number of turbines located within the cell. The Fitch et al. (2012) WFP is coupled to the MYNN PBL scheme in the standard WRF release (including the bug fix of Archer et al., 2020), allowing for direct comparisons with the 3D PBL implementation.

The initial work of Juliano et al. (2022) and Rybchuk et al. (2022) focused on developing and testing the 3D PBL scheme in idealized model configurations, mostly with flat terrain or over open water. Juliano et al. (2022) considered idealized convective boundary layer and sea breeze tests, as well as a mountain-valley test with simple terrain, while Rybchuk et al. (2022) considered the offshore environment. Arthur et al. (2022) and Wiersema et al. (2023) subsequently evaluated 3D PBL performance relative to standard WRF options in real complex-terrain scenarios. However, further testing of the model is necessary to ensure its robustness.

With this in mind, the present work has two main goals. The first is to evaluate the 3D PBL scheme in a complex-terrain region that is relevant to wind energy. The second is to build on the work of Rybchuk et al. (2022) by testing the WFP coupled to the 3D PBL scheme in a realistic configuration with terrain. Ultimately, this work aims to better establish the utility of the 3D PBL scheme for wind energy applications.

## 2 Data and methods

### 2.1 Case study and observational data

The Altamont Pass Wind Resource Area (APWRA) is a collection of wind plants located in a gap within the Diablo Range of north-central California. The gap is just east of San Francisco Bay and south of the San Francisco Bay Delta, and is roughly bounded by Mt. Diablo to the northwest and the greater Diablo Range to the southeast (see Figure 1). With nearly 200 turbines and roughly 326 MW of installed capacity spread over six plants (excluding very small, old 65 kW turbines), the APWRA is the fifth largest wind energy installation in California and one of the oldest commercial wind farms in the United States, with the first turbines installed in 1981 (see Hoen et al., 2018). The typical annual cycle of wind energy production in the APWRA is shown in Figure 2 in terms of monthly capacity factors, defined as the ratio of actual production to the maximum possible production (i.e., if all turbines operated at full capacity) during each month.

The turbines in the APWRA are especially productive over the summer months when a synoptic pressure difference between the ocean and the land drives westerly/southwesterly winds that are channeled through the Altamont Pass (see, e.g., Zaremba and Carroll, 1999). These winds are modulated by diurnal temperature variability, which enhances the land-sea pressure difference, leading to peak wind speeds in the late afternoon to early evening local time (see, e.g., Wharton et al., 2015). The

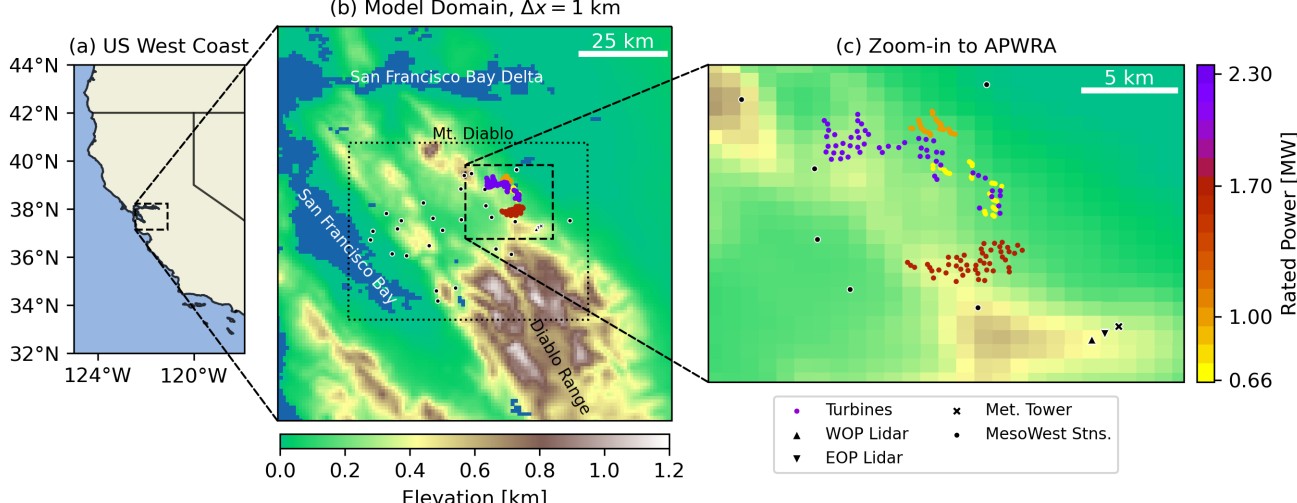

**Figure 1.** A map of the study region, zooming in from (a) the US west coast, to (b) the WRF model domain, to (c) the APWRA. Included in (b) and (c) are the locations of observation stations (black symbols) used for model evaluation, the locations of APWRA wind turbines at the time of the HilFlowS study (colored by their rated power), and terrain elevation as represented in the model, with water shown in blue. Dashed-line boxes indicate zoomed-in regions in the next panel to the right, while the dotted-line box in (b) indicates the region shown in Fig. 7.

regularity of the summertime speedup events, combined with the importance of terrain-induced wind acceleration, makes them a useful case study for evaluating mesoscale models (see, e.g., Banta et al., 2020, 2023).

The Hill Flow Study (HilFlowS; Wharton and Foster, 2022) consisted of two vertically profiling ZephIR300 lidars and a 52-m meteorological tower deployed at Lawrence Livermore National Laboratory Site 300, roughly 10 km southeast of the APWRA wind plants, during the mid-to-late summer of 2019. HilFlowS was conducted along three parallel ridgelines that run northwesterly to southeasterly in the Diablo Range, making them perpendicular to the predominant summertime, southwesterly (onshore) wind direction. Lidars were deployed on the first two (upwind) parallel ridgelines at the Western Observation Point

(WOP; Atmosphere to Electrons, 2019c) and Eastern Observation Point (EOP; Atmosphere to Electrons, 2019b), which are separated by a line-of-sight distance of 860 m. The WOP ridgeline has a higher peak (527 m MSL), while the EOP peak is slightly lower (448 m MSL). The ridgeline slopes, respectively, are 22° and 13° along the predominant wind direction of 240°. The meteorological tower (Atmosphere to Electrons, 2019a) is found on the third ridgeline and is at an elevation of 395 m MSL. The study area and surrounding region is largely covered by grassland. All instrument and turbine locations are included in

Figure 1.

Wind speed data from the two lidars are used here to evaluate model performance between the surface and 150 m AGL, spanning the vertical range of the turbines in the APWRA. Both lidars gathered horizontal wind speed, wind direction, and vertical velocity data at 10, 20, 30, 38, 50, 60, 70, 80, 90, 120, and 150 m AGL (note that 38 m is a fixed calibration height),

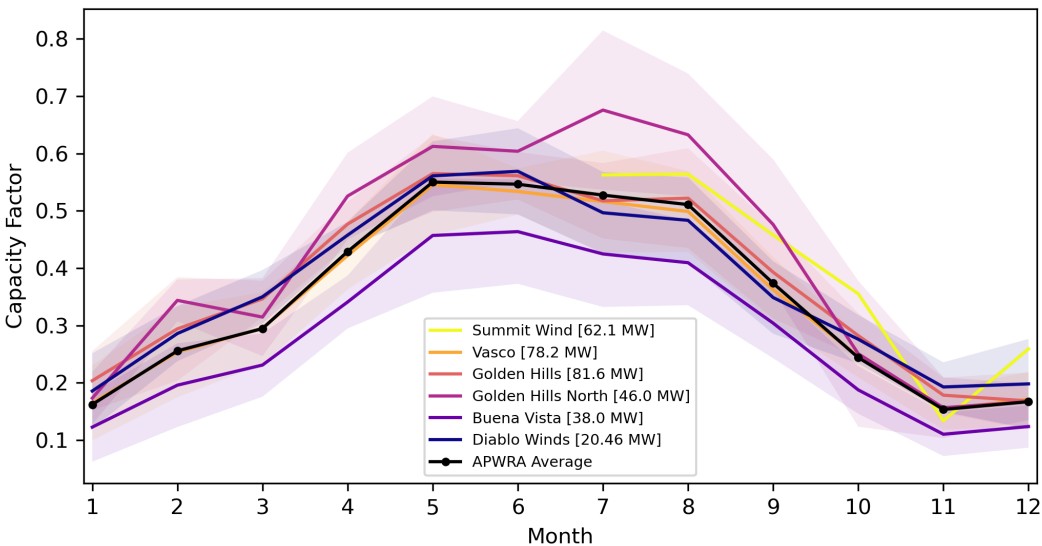

**Figure 2.** Monthly capacity factors for the six wind plants in the APWRA, based on EIA-reported data (EIA, 2023a, b) averaged over 2014–2021. The shaded area represents ±1 standard deviation. The average over all plants is weighted by plant capacity as noted in the legend. Note that Summit Wind became operational in 2021.

between 9 July and 23 September 2019. Horizontal wind speed, direction, air temperature, and air pressure data are also
available at 1 m AGL from an on-board meteorological station, although only the wind speed and direction data are used here.

While the lidars completed their scan strategy roughly once every 15 s, the data have been averaged in 10-min intervals as in Wharton and Foster (2022). Over the study period, the WOP lidar had greater than 98% data availability for horizontal wind speed/direction, and roughly 90% data availability for vertical velocity. The EOP lidar ran on solar/battery power, which resulted in slightly lower data availability of roughly 84% and 77%, respectively. Lower data availability for the vertical
velocity, relative to the horizontal, is a result of standard quality control filtering applied by the lidars when calculating 10-min averages, which removes the vertical velocity when rain or fog are detected. Diurnal composite averages over the nearly 3-month-long data record were analyzed by Wharton and Foster (2022) and shown to be robust; a similar composite-averaging approach is used in the present study for model evaluation.

Horizontal wind speed, wind direction, and vertical velocity are calculated from lidar observations using the velocity-azimuth
display (VAD) technique for each measurement height. Note that the ZephIR300 does not have a vertically pointing beam, thus vertical velocities are not measured directly. TKE is calculated using high-frequency variance measurements during post-processing (see section 3.1.2). Reported accuracy for the ZephIR300 in ideal site conditions (e.g., flat, homogeneous terrain) is ±0.25% for wind speed and direction. However, the HilFlowS experiment was not conducted under these ideal conditions. In hilly terrain, assumptions about the horizontal homogeneity of the flow across the lidar's observation volume may be invalid,

leading to errors in the measured horizontal wind speed as large as $\pm 10\%$ (Bingöl et al., 2009). Although Bingöl et al. (2009) did not quantify errors in vertical velocities, these are also expected to be present in complex terrain due to the ZephIR300 lidar's lack of a vertically pointing beam.

An earlier experiment in the APWRA (Wharton et al., 2015) that used identical ZephIR300 lidars to measure hill speedup flows and their effects on power production assessed terrain-induced measurement errors with the Dynamics software package

provided by ZephIR Ltd. As discussed therein, the software converts raw lidar line-of-sight velocity data into unbiased measurements of wind speed and wind direction for hilly sites, based on the work of Bingöl et al. (2009). In Wharton et al. (2015), conversion factors for all wind directions and measurement heights ranged from +1% to +8% for the hill lidar, within the range of the Bingöl et al. (2009) study. Moreover, the correction factors associated with the predominant wind direction were closer to zero: +3% for the hill lidar and -2% for the base lidar near the bottom of the hill.

The conversion factors in Wharton et al. (2015) were calculated for a hill that is similar to those at the HilFlowS site, and are presented here for additional context. However, conversion factors are not recalculated for the present study. Rather, the potential $\pm 10\%$ calculated by Bingöl et al. (2009) is used to conservatively bound the potential mean error in the measured horizontal wind speed. It should be noted that prior to the HilFlowS experiment, the lidars were cross-compared with high agreement (see Wharton and Foster, 2022), providing confidence in their use for model evaluation.

To supplement lidar observations, wind speed and temperature data are available from the meteorological tower at 10, 23, and 52 m AGL. Wharton and Foster (2022) used these data to assess atmospheric stability via the bulk Richardson number; here, the temperature data are used for model evaluation. Furthermore, before the start of HilFlowS, the lidars were deployed at the base of the meteorological tower to assess instrument agreement. That dataset showed strong agreement between the lidars and the tower, with r-squared values of 0.97-0.99 for all measurement levels.

To further examine the spatial variability of model performance, 10-m wind speed data from nearby surface meteorological stations in the MesoWest network (Mesonet, 2023) are used. Although proprietary turbine data from the APWRA wind plants are not generally available, public power production data reported to the United States Energy Information Administration (EIA) on a monthly basis (EIA, 2023a, b) are used to evaluate estimates of wind power production from the WFP. Note that site-specific wind power studies have been performed previously in the APWRA, as presented in Wharton et al. (2015) and

Bulaevskaya et al. (2015).

Rios et al. (2025) used HilFlowS lidar data to evaluate the aforementioned HRRR model, which is used frequently for forecasting within the wind energy industry (Shaw et al., 2019). Rios et al. (2025) found that while HRRR captured the general diurnal trend of the observed speedup events, it overestimated hub-height wind speeds (by as much as 3 m s$^{-1}$) during nighttime hours, and underestimated hub-height wind speeds by as much as 2 m s$^{-1}$ during daytime hours. Wind speed errors

also varied spatially and as a function of the predominant wind direction associated with different synoptic conditions. These results serve as a baseline for the present study, which explores the effects of increased grid resolution (relative to HRRR) and PBL treatment on model performance.

## 2.2 Model configuration

### 2.2.1 Domain and model options

The WRF model version 4.4 is employed with a horizontal grid spacing of 1 km over the 120×120 km domain depicted in Figure 1b. The model is initialized on 6 July 2019 0000 UTC, allowing for roughly two days of spinup time prior to observational comparisons, and run through 24 September 2019 0000 UTC. Initial and boundary conditions are derived from hourly HRRR analysis fields (at the 0th forecast hour), but interior nudging is not employed due to the relatively small domain. The WRF namelist and wind turbine specification files used in this study are archived under Arthur (2024).

Simulations are completed with two model configurations, varying only the treatment of SGS turbulent mixing. The first configuration is treated as a control and roughly corresponds to the standard HRRR setup, while the second configuration employs the 3D PBL scheme. Recall that HRRR uses a horizontal grid spacing of 3 km; the present value of 1 km was chosen to increase resolution relative to HRRR while also approaching both the upper limit of traditional mesoscale models and the lower limit of the turbulence gray zone.

In the control configuration, vertical turbulent mixing is treated using the MYNN level 2.5 PBL scheme ($bl\_pbl\_opt = 5$), while horizontal mixing is not treated explicitly; rather, horizontal smoothing is employed with WRF's 2D Smagorinsky scheme ($km\_opt = 4$). In the second configuration, both vertical and horizontal turbulent mixing are treated using the 3D PBL scheme. In both configurations, local curvilinear-grid metric terms are used in the calculation of horizontal gradients (as with WRF's $diff\_opt = 2$), although $diff\_opt$ is set to 0 when the 3D PBL scheme is used. All other model options are identical

between the two configurations.

    Note that following Rybchuk et al. (2022), Arthur et al. (2022), and Wiersema et al. (2023), the PBL approximation (Mellor, 1973; Mellor and Yamada, 1982) is used within the 3D PBL scheme ($pbl3d\_opt = 1$) to improve computational efficiency and numerical stability (see discussions therein, and in Juliano et al., 2022). Indeed, the full 3D PBL scheme was found to be computationally unstable in the present domain, likely due to the turbulence length-scale calculation. This was also the case in

the complex-terrain studies of Arthur et al. (2022) and Wiersema et al. (2023). With the PBL approximation, the divergences of horizontal turbulence shear stresses and turbulent fluxes are retained in the prognostic equations for momentum and scalars, respectively. However, horizontal gradients are neglected in the system of equations used to calculate the stresses and fluxes, allowing them to be determined analytically. Horizontal gradients are also neglected in the prognostic equation for TKE. Thus, TKE production due to horizontal shear, which has been found by previous studies to be important in complex terrain (Zhong

and Chow, 2012; Muñoz-Esparza et al., 2016; Goger et al., 2018), is not considered here. Potential ramifications of using the PBL approximation in this study are discussed further below.

    For consistency with the HRRR forcing, the present model runs use the HRRR atmospheric physics suite following Benjamin et al. (2016). This includes the Rapid Update Cycle (RUC) land-surface model ($sf\_surface\_physics = 3$), the Thompson aerosol-aware microphysics scheme ($mp\_physics = 28$; Thompson, 2014), and the RRTMG radiation schemes ($ra\_sw\_physics = 4$ and $ra\_lw\_physics = 4$; Iacono et al., 2008). However, for compatibility with the 3D PBL scheme, the revised MM5 surface

layer scheme ($sf\_sfclay\_physics = 1$) is used instead of the MYNN scheme ($sf\_sfclay\_physics = 5$). Additionally, fol-

lowing Arthur et al. (2022), WRF's option to add positive-definite 6th-order horizontal diffusion ($diff\_6th\_opt = 2$) is used in both configurations with a factor of 0.25. The added diffusion is purely numerical and is used to damp grid-scale noise. However, to prevent over-diffusion in regions of sloping terrain, where numerical diffusion is already expected to be relatively large, the added 6th-order diffusion is linearly damped between slopes of 0 and 0.05 (2.86°) and turned off for larger slopes (using the namelist options $diff\_6th\_slopeopt = 1$ and $diff\_6th\_thresh = 0.05$).

The vertical grid spacing is modified from HRRR in the present study to increase vertical grid resolution within the turbine layer. HRRR uses 50 vertical levels, with a vertical grid spacing of $\Delta z \approx 16$ m at the surface such that the first half level (the lowest level at which temperature and velocities are calculated) is located at roughly 8 m AGL. The vertical grid spacing is stretched above the surface, as detailed in Benjamin et al. (2016), with a domain top of roughly 25 km. Here, $\Delta z$ is held constant at 16 m between the surface and roughly 300 m AGL (19 levels), and stretched with a factor of 1.1 above, with a total of 69 levels. Although Tomaszewski and Lundquist (2020) and Rybchuk et al. (2022) recommend setting $\Delta z$ to 10 m or less with the WFP, this was found to be computationally unstable for the 3D PBL run; ongoing improvements to the 3D PBL scheme may alleviate this issue in the future. Note also that the present model runs use WRF's standard terrain-following vertical coordinate system ($hybrid\_opt = 0$), as in Arthur et al. (2022). Although WRF's hybrid vertical coordinate ($hybrid\_opt = 1$) is used in HRRR version 3 (used here for model forcing, see Dowell et al., 2022), the hybrid coordinate system primarily affects predictions above the boundary layer and is therefore not considered here.

### 2.2.2 Wind turbine representation

The Fitch et al. (2012) WFP, including the bug fix of Archer et al. (2020), is used in both model runs to predict the power output by APWRA turbines during the study period. Turbines are represented in the WFP by their location, hub height, rotor diameter, and power/thrust curves. The necessary WRF-WFP input files used in this study are archived under Arthur (2024). For consistency with Rybchuk et al. (2022), the wind farm TKE factor (WRF namelist variable $windfarm\_tke\_factor$), which controls the amount of TKE added to the flow, is set to 1. This differs from the value of 0.25 used by Archer et al. (2020). As of the time of this writing, there is no clear consensus in the literature on the optimal choice for this parameter (Larsén and Fischereit, 2021; Ali et al., 2023). Note that although wind farm wake dynamics are predicted by the WFP, they are not a focus of the present study. Moreover, wakes are not expected to reach the HilFlowS observation sites given the complex terrain and predominant wind direction of 240°.

At the time of the study period, the APWRA consisted of 171 total turbines spread across 5 wind plants, summarized in Table 1. Turbine locations (as shown in Figures 1 and 7) and specifications are extracted from the United States Wind Turbine Database (Hoen et al., 2018). However, the present analysis excludes very small (65 kW), old turbines that are still listed in Hoen et al. (2018).

Because the power and thrust curves for the actual APWRA turbines are generally proprietary, comparable publicly available curves are used here (see Table 1). The General Electric (GE) 2.3, Siemens 2.3, and GE 1.7 MW APWRA turbines are matched as closely as possible to the generic dataset of NREL (2022), which is based on the OpenFAST model (https://github.com/OpenFAST) and includes both power and thrust curves. However, since lower-power turbines are not included in the NREL (2022) dataset,

**Table 1.** A summary of wind plants in the APWRA during the summer 2019 study period. Actual turbine specifications are based on Hoen et al. (2018), while modeled specifications are based on the best-available public data as described in the text. Turbines are listed in terms of the manufacturer (Mfr), the rated power $P_R$ (see colors in Figure 1), the hub height $H$, and the rotor diameter $D$. The manufacturer is listed as "NREL" when the generic dataset of NREL (2022) is used. Note that the 62 MW Summit Wind plant shown in Figure 2 was installed after the study period and is therefore not included here. The Patterson Pass and Patterson Wind plants (included in Hoen et al., 2018), which consist of very small (65 kW), old turbines, are also not considered in the analysis.

| Wind Plant | # Turbines | Actual | | Modeled | |
|---|---|---|---|---|---|
| | | Mfr-$P_R$ [MW] | $H, D$ [m] | Mfr-$P_R$ [MW] | $H, D$ [m] |
| Golden Hills North | 20 | GE-2.3 | 80, 116 | NREL-2.3 | 80, 116 |
| Vasco | 34 | Siemens-2.3 | 80, 101 | NREL-2.3 | 80, 107 |
| Golden Hills | 48 | GE-1.7 | 80, 100 | NREL-1.7 | 80, 103 |
| Buena Vista | 38 | Mitsubishi-1.0 | 55, 61 | Bonus-1.0 | 55, 54 |
| Diablo Winds | 31 | Vestas-0.66 | 60, 47 | Vestas-0.66 | 60, 47 |
| Total | 171 | 264.26 MW | | 264.26 MW | |

additional curves are gathered from the dataset of wind-turbine-models.com (2024b, a). Within this dataset, a power curve for the Vestas 0.66 MW turbine is available (wind-turbine-models.com, 2024b); however, the thrust curve must be interpolated from the generic NREL 1.7 model. For the Mitsubishi 1.0 MW turbine, a comparable power curve from a Bonus 1.0 MW turbine (wind-turbine-models.com, 2024a) is used, again with a thrust curve interpolated from the generic NREL 1.7 model.

The modeled APWRA turbines have the same total rated capacity of 264.24 MW as the installed turbines at the time of the study period (Table 1). Furthermore, Siedersleben et al. (2020) demonstrated that the exact details of the power and thrust curves are not critical to WFP performance. Ultimately, modeled capacity factors, rather than raw power production estimates, are presented below. Thus, the effect of differences between the actual and modeled turbine specifications is expected to be small.

## 3   Model evaluation

### 3.1   Vertical variability

#### 3.1.1   Wind speed, wind direction, and temperature

Model performance is first evaluated through comparison to lidar observations from the HilFlowS experiment (Wharton and Foster, 2022). The instantaneous model error $E_{VAR}$ is defined as

$$E_{VAR} = VAR_{WRF} - VAR_{OBS}, \tag{1}$$

where $VAR$ is the meteorological variable, either horizontal wind speed $V$, wind direction $\phi$, or vertical velocity $w$. The error is calculated at 10-min intervals, corresponding to the frequency of the processed lidar data as well as the model output. This calculation requires spatial interpolation of the model data to the lidar measurement locations. Model data are first interpolated horizontally to the latitude/longitude of the lidar, using nearest neighbor interpolation, and are then linearly interpolated to the lidar vertical levels. Although several figures herein present observed wind speed and direction from the lidar's on-board meteorological station at 1 m AGL, model errors are not evaluated at this height because extrapolation below the first half level (at roughly 8 m AGL) would be required. Additionally, note that $E_\phi$ is adjusted to account for the cyclical nature of the wind direction: if the raw $E_\phi$ value is less than $-180°$ (greater than $180°$), it is adjusted by $+360°$ ($-360°$).

Due to the day-to-day consistency of the observed speedup events, diurnal composite averages are used to summarize model performance over the nearly 3-month-long study period (see Figure 3). The diurnal composite bias is therefore defined as

$$B_{VAR} = \langle VAR_{WRF} - VAR_{OBS} \rangle_C = \langle E_{VAR} \rangle_C, \tag{2}$$

where the angle brackets denote a time average, in this case a diurnal composite denoted by the subscript $C$. A positive bias indicates an overestimate by the model, while a negative bias indicates an underestimate. Composite averages are performed between 9 July 2019 0000 PST and 23 September 2019 0000 PST such that only complete days in local time (PST=UTC-8) are included in the analysis. Model results in Figure 3 are shown for the 3D PBL configuration, although those for the MYNN configuration are visually similar; more detailed comparisons between the two are discussed below. Note that while the figures in this section are shown at the WOP site for brevity, the discussion generally applies to both sites unless otherwise noted. A selection of time-height averaged error metrics are shown for both sites in Table 2.

As presented in Wharton and Foster (2022), observed winds at the study site begin to accelerate around midday, reaching a peak between 1500–2100 PST. Winds then decelerate over the course of the night, reaching a minimum between 0600–0900 PST (Figure 3a). The speedup flows, which are channeled through the Altamont Pass, are predominantly southwesterly (230-250°), while daytime flows are more variable (Figure 3c). The speedup flows at the study site are associated with subsidence, a negative vertical velocity (blue colors in Figure 3e), and increased horizontal wind speeds near the surface (yellow colors in Figure 3a). This suggests that vertical convergence leads to horizontal divergence and an acceleration of the flow near the surface.

While the model captures the timing and direction of the speedup flows well (Figure 3b,d), wind speeds are generally overestimated above 30 m AGL, especially between 0000–0300 PST (red colors in Figure 3b). Conversely, wind speeds are underestimated near the surface, indicating that the model fails to capture near-surface accelerations. This highlights an inherent limitation of the vertical grid setup, which, although finer than HRRR, has only 1–2 model levels ($\Delta z \approx 16$ m) within the observed jet-like flow layer below roughly 30 m AGL. While the model captures some negative vertical velocities at the study site during the speedup events (see contours in Figure 3f), its vertical velocities are too weak and thus do not translate to near-surface accelerations of the magnitude seen in the observations.

Several time-height average error metrics (following e.g., Chang and Hanna, 2004; Smith et al., 2018; Wiersema et al., 2020; Arthur et al., 2022) are used to compare the performance of the two model configurations over the course of the study period.

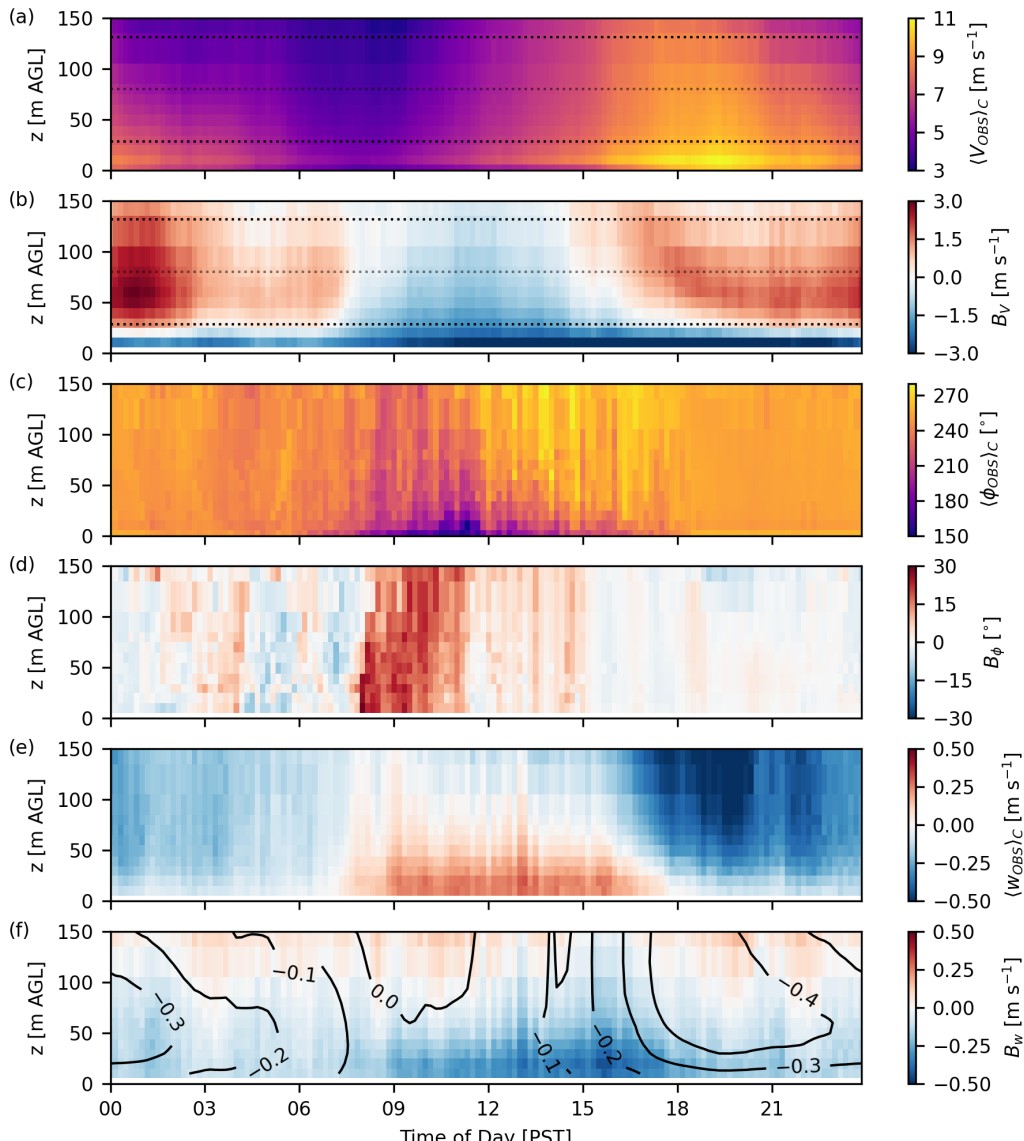

**Figure 3.** Diurnal composite-average WOP lidar observations and 3D PBL model bias. Positive bias (red) indicates an overestimate by the model, while negative bias (blue) indicates an underestimate. Shown are wind speed $V$ (a,b), wind direction $\phi$ (c,d), and vertical velocity $w$ (e,f). Note that (a) and (c) include data from the lidar's on-board meteorological station at 1 m AGL, but model errors are not evaluated at this height. To contextualize the vertical velocity bias in (f), contour lines are shown for the modeled vertical velocity $\langle w_{WRF} \rangle_C$ in 0.1 m s$^{-1}$ increments. Dotted lines in (a) and (b) indicate the rotor-swept area of the most prevalent generic turbine model in the simulations, with hub-height $H = 80$ m and rotor diameter $D = 103$ m (NREL-1.7; see Table 1).

**Table 2.** Error metrics, as defined in Equations 3, 4, and 5, for each model configuration at each lidar site. Metrics are time averaged over the full study period and vertically averaged over two separate layers: the rotor layer (lidar measurement heights of 30, 38, 50, 60, 70, 80, 90, and 120 m AGL) and a near-surface layer below the rotor layer (lidar measurement heights of 10 and 20 m AGL). The rotor layer is based on the most prevalent generic turbine model in the simulations, with hub-height $H = 80$ m and rotor diameter $D = 103$ m (NREL-1.7; see Table 1).

| Site | WOP | WOP | WOP | WOP | EOP | EOP | EOP | EOP |
|---|---|---|---|---|---|---|---|---|
| Vertical Layer | Near-Surface | Near-Surface | Rotor | Rotor | Near-Surface | Near-Surface | Rotor | Rotor |
| Model | 3D PBL | MYNN | 3D PBL | MYNN | 3D PBL | MYNN | 3D PBL | MYNN |
| $FB_V$ | -0.29 | -0.28 | 0.069 | 0.059 | -0.34 | -0.33 | 0.034 | 0.024 |
| $NMAE_V$ | 0.35 | 0.34 | 0.25 | 0.25 | 0.40 | 0.39 | 0.26 | 0.25 |
| $SAA$ [°] | 12 | 12 | 12 | 12 | 13 | 13 | 12 | 12 |
| $FB_w$ | -0.98 | -0.96 | -0.32 | -0.30 | -0.15 | -0.14 | 0.24 | 0.25 |
| $NMAE_w$ | 1.2 | 1.2 | 0.94 | 0.94 | 0.69 | 0.68 | 0.73 | 0.73 |

The fractional bias is defined as

$$FB_{VAR} = \frac{\left\langle \overline{E_{VAR}} \right\rangle}{0.5 \left( \left\langle \overline{|VAR_{WRF}|} \right\rangle + \left\langle \overline{|VAR_{OBS}|} \right\rangle \right)}, \tag{3}$$

and the normalized mean absolute error is defined as

$$NMAE_{VAR} = \frac{\left\langle \overline{|E_{VAR}|} \right\rangle}{0.5 \left( \left\langle \overline{|VAR_{WRF}|} \right\rangle + \left\langle \overline{|VAR_{OBS}|} \right\rangle \right)}, \tag{4}$$

where angle brackets denote a time average over 9 July 2019 0000 PST through 23 September 2019 0000 PST and the overbar denotes a vertical average over available lidar measurement heights. Note that the absolute value operation in the denominator is only relevant for the vertical velocity, which has both positive and negative values; the horizontal wind speed is positive by definition.

For the wind direction, the scaled average angle is defined as

$$SAA = \frac{1}{N \left\langle \overline{V_{WRF}} \right\rangle} \sum_{i=1}^{N} V_{WRF,i} |E_{\phi,i}|, \tag{5}$$

where $N$ is the total number of observations (in both time and height) for the given lidar. $SAA$ weighs wind direction errors based on the modeled wind speed at the given observation location and time, assuming that directional errors at low wind speeds are less impactful.

Error metrics are summarized in Table 2 for both model configurations and lidar sites. The metrics shown in the table are time averaged over the full study period and vertically averaged over two separate layers: the rotor layer (lidar measurement

heights of 30, 38, 50, 60, 70, 80, 90, and 120 m AGL) and a near-surface layer below the rotor layer (lidar measurement heights of 10 and 20 m AGL). The rotor layer is based on the most prevalent generic turbine model in the simulations, with hub-height $H = 80$ m and rotor diameter $D = 103$ m (NREL-1.7; see Table 1). Overall, error metrics are nearly equal for the MYNN and 3D PBL configurations, with a slight overestimate of the wind speed in the rotor layer and a larger underestimate near the surface.

Model performance is examined in more detail using composite-average wind speed profiles, presented in Figure 4. During the onset of the speedup events, the 3D PBL configuration predicts faster wind speeds than the MYNN configuration throughout the lidar range, showing reduced negative bias compared to the observations, especially below hub height (assumed to be 80 m; Figure 4, 1200–1500 PST). This may be due to slightly improved predictions of vertical mixing of higher momentum from aloft; during this time, prior to jet development, the winds follow a standard quasi-logarithmic profile. The 3D PBL scheme has been shown previously, in idealized tests, to improve model performance during daytime convective conditions (Juliano et al., 2022).

During the peak of the speedup flow, however, the 3D PBL configuration begins to overestimate wind speeds below hub height, showing a slightly more pronounced jet near the surface relative to the MYNN configuration (Figure 4, 1800–2100 PST). This pronounced jet persists into the night for both model configurations, until roughly 0000 PST. Then, as the flow decelerates in the early morning, both model configurations tend to overestimate wind speeds throughout the rotor layer (Figure 4, 0300–0600 PST). Finally, when the flow reaches a minimum around 0900 PST, both models underestimate wind speeds throughout the rotor layer, with a slightly larger underestimate in the 3D PBL configuration.

To expand upon the composite-average wind speed analysis in Figure 4, results from a sample day during the study period, 21 July 2019, are presented in Appendix A. This day was chosen to highlight differences between the 3D PBL and MYNN configurations while also showing consistency with the composite-average results. The same day was highlighted in the original HilFlowS study (Wharton et al., 2015, see Figure 5 therein). The reader is referred to Appendix A for additional discussion.

Taken together, wind speed error metrics (Table 2), composite-average profiles (Figures 3 and 4), and results from the sample day (Figure A1) suggest that for both model configurations, the predicted amount of vertical mixing is inadequate to transport higher momentum downward from aloft. This results in a persistent negative wind speed bias below roughly 30 m AGL throughout the day. During speedup events, too much momentum remains within the rotor layer. Although both model configurations produce a pronounced jet below hub-height and reduced wind speeds above (Figure 4, 2100–0600 PST), wind speeds are generally overestimated in the rotor layer and underestimated near the surface.

Further development and testing of the 3D PBL scheme could lead to more accurate wind speed predictions, especially if near-surface vertical resolution is increased. Notably, the 3D PBL scheme allows more run-time flexibility in turbulence treatment (via, e.g., the closure constants) relative to MYNN and other 1D PBL schemes, which could facilitate improved predictions of vertical mixing. However, as mentioned previously, the 1 km horizontal grid spacing of the present simulations inherently limits the ability of the model to capture the observed flows. In particular, the hilly topography of the HilFlowS site, including the individual ridgelines on which the lidars were deployed, is not fully captured (see Figure 1).

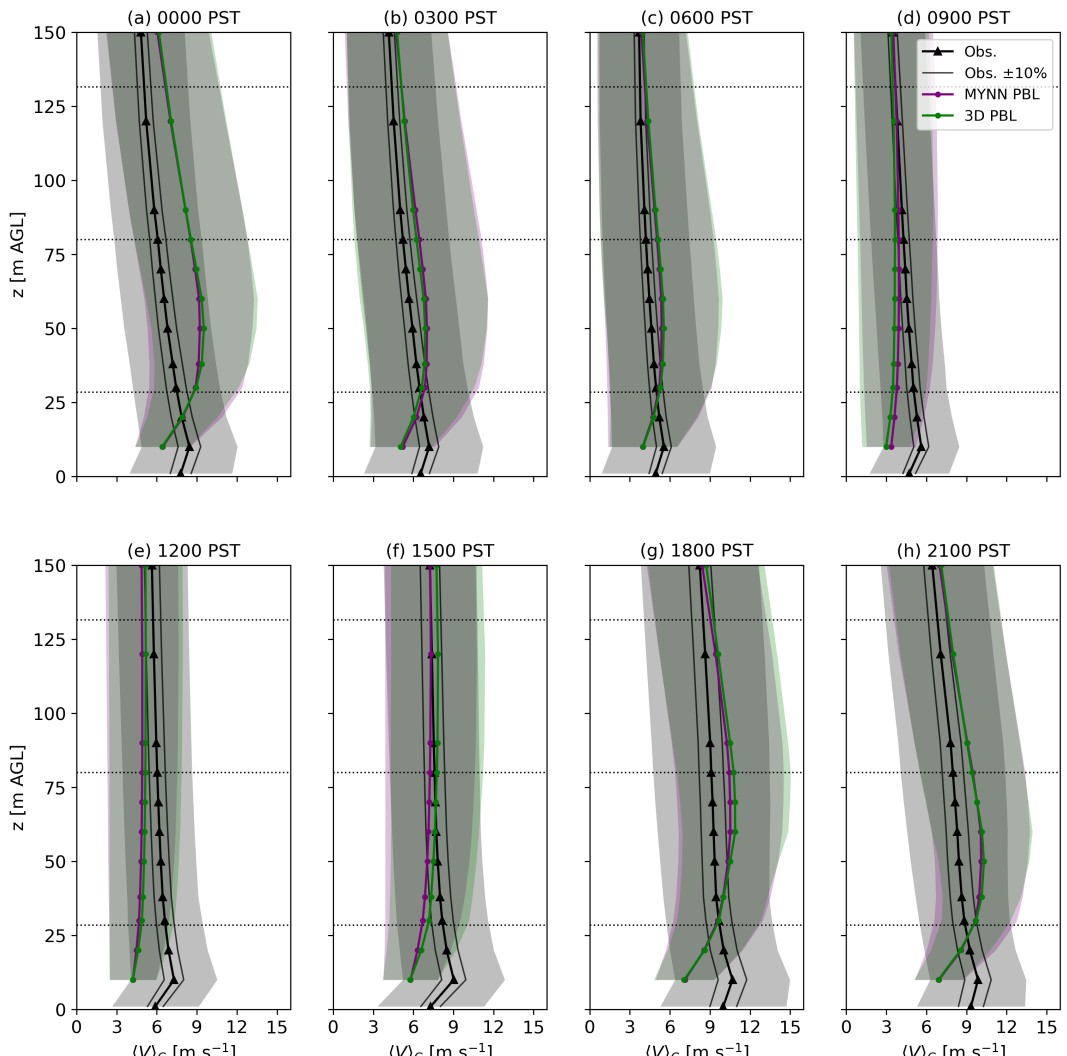

**Figure 4.** Diurnal composite-average wind speed profiles, shown for WOP lidar observations and both model configurations. Potential mean error bounds of ±10% are also shown for the lidar observations following Bingöl et al. (2009). Profiles are averaged over the hour indicated at the top of each panel, and model data have been interpolated to the vertical levels of the lidar, as in Figure 3. Note that data is included from the lidar's on-board meteorological station at 1 m AGL, but model errors are not evaluated at this height. The shaded regions show ±1 standard deviation over the given hour of the diurnal composite. Dotted lines indicate the rotor-swept area of the most prevalent generic turbine model in the simulations, with hub-height $H = 80$ m and rotor diameter $D = 103$ m (NREL-1.7; see Table 1).

To further contextualize model wind speed biases, it is important to recall (see Section 2.1) that conically scanning lidars such as the ZephIR300 deployed during HilFlowS are known to experience errors in complex terrain. These errors result from

violating the assumption of homogeneity that the lidars use to deduce the horizontal and vertical wind speeds. In particular, Bingöl et al. (2009) found horizontal wind speed errors as large as 10%, while Wharton et al. (2015) found comparable or smaller values for a similar site in the APWRA. As a conservative estimate, the findings of Bingöl et al. (2009) imply mean horizontal wind speed errors as large as roughly 1.5 m s$^{-1}$ in the HilFlowS lidar observations (see gray bounding lines in Figure 4). In general, the expected maximum lidar error is smaller than the model bias, especially near the surface. Thus, the potential lidar error is not expected to affect the present conclusions related to model evaluation.

To complement wind profile comparisons at the lidar sites, temperature profiles at the meteorological tower site are shown in Figure 5. Note that the meteorological tower is on a similar hill to that found at WOP and EOP, and is separated by a line-of-sight distance of 950 m from EOP. The 3D PBL configuration shows slightly better agreement with the observed temperature profile for most hours of the day, especially during daytime conditions when the vertical temperature gradient is negative (0900–1500 PST). This time corresponds to reduced wind speed bias at both lidar sites. Small improvements in the temperature prediction are also seen during the evening transition, as the vertical temperature gradient becomes positive (1800–2100 PST). At this time, the 3D PBL scheme produces a more pronounced near-surface jet, but shows larger wind speed bias relative to MYNN, as discussed above.

### 3.1.2 Turbulence kinetic energy

Both the 3D PBL and MYNN schemes parameterize SGS turbulence shear stresses and turbulent fluxes using a prognostic equation for the SGS TKE. Thus, TKE predictions can provide insights into model performance. TKE estimates are also available from the HilFlowS lidars, and are calculated as

$$TKE = \frac{1}{2} \left( \langle u'^2 \rangle + \langle v'^2 \rangle + \langle w'^2 \rangle \right),$$ (6)

where $u$, $v$, and $w$ denote velocities in the zonal, meridional, and vertical directions, respectively, and brackets denote 10-min averages. Perturbation quantities, denoted by the prime symbol, are calculated as the difference between the high-frequency (15-s) time series and a detrended time series based on 2-min averages.

Note that both the observed and modeled TKE values have inherent limitations. The lidar TKE estimates are spatially averaged over the lidar's conical scanning volume and are time-averaged in 10-min windows. Furthermore, the estimated TKE is limited by the 15-s sampling frequency (see additional discussion in Sathe et al., 2011). Lidar TKE estimates are also influenced by complex terrain, as discussed above for wind speeds. The modeled TKE is fully parameterized (i.e., it is assumed that there is no resolved TKE) in each model grid cell and is output as an instantaneous value every 10 min. Ultimately, these limitations preclude direct comparison of observed and modeled TKE values (i.e., bias calculations). In what follows, the time-height structure of the TKE is compared qualitatively between the observations and the model, while only the modeled TKE values are compared quantitatively.

Observed and modeled TKE profiles are shown in Figure 6 for the WOP site. In the midday, observed TKE is elevated throughout the lidar's vertical range due to surface heating and associated atmospheric instability. The speedup flows are also accelerating during this time, leading to peak TKE values below 50 m AGL due to shear associated with the jet-like velocity

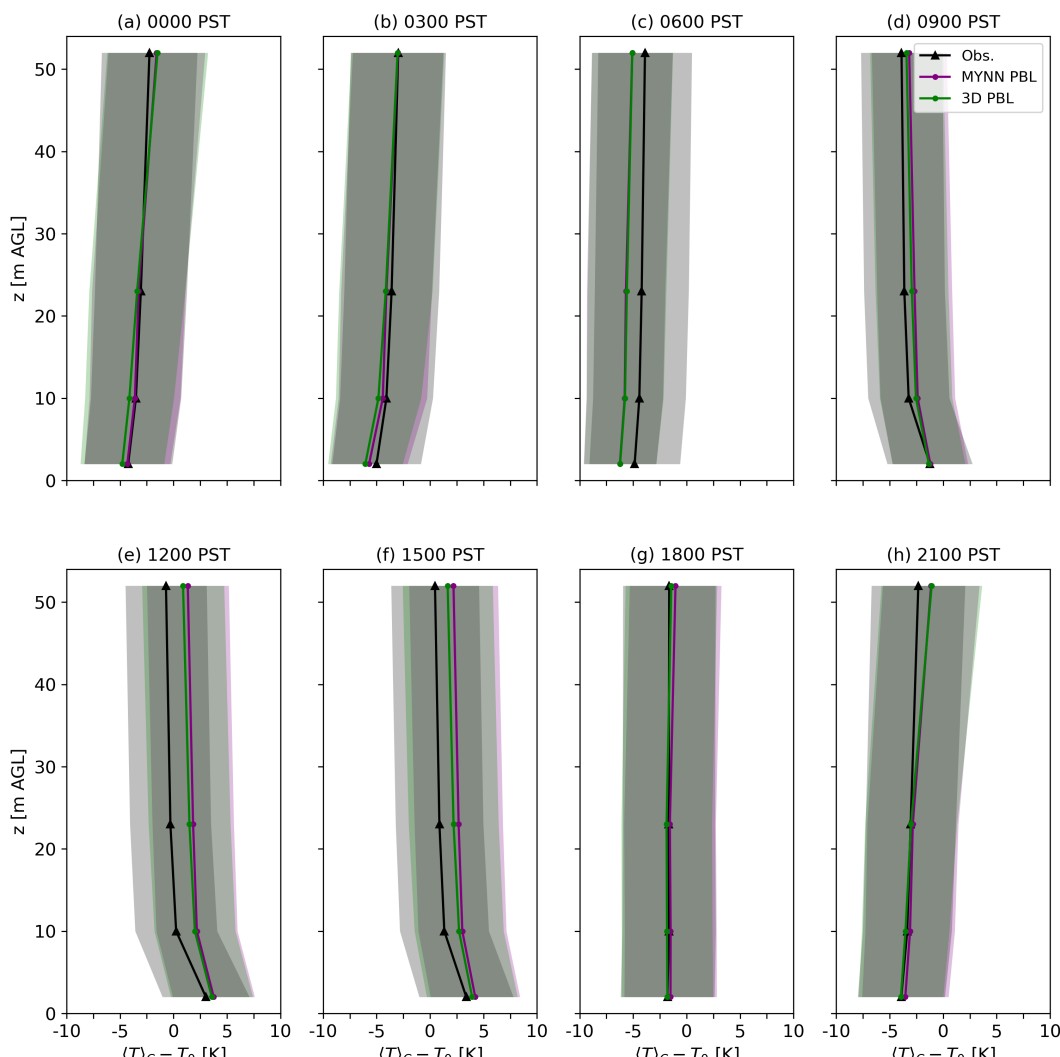

**Figure 5.** Diurnal composite-average temperature profiles ($T_0 = 300$ K), shown for the HilFlowS 52-m meteorological tower and both model configurations. Profiles are averaged over the hour indicated at the top of each panel, and model data have been interpolated to the vertical levels of the tower observations. The shaded regions show $\pm 1$ standard deviation over the given hour of the diurnal composite. Note that the vertical axis range is limited to the tower height.

profile (Figure 6a, 1200–1800 PST). Both model configurations capture elevated TKE during this time (Figure 6b,c). However, the MYNN configuration generally predicts larger TKE values relative to the 3D PBL configuration. This is likely because the 3D PBL scheme with the PBL approximation introduces additional horizontal mixing, relative to MYNN, without added TKE production due to horizontal shear. Reduced TKE in the 3D PBL configuration is associated with improved velocity profile

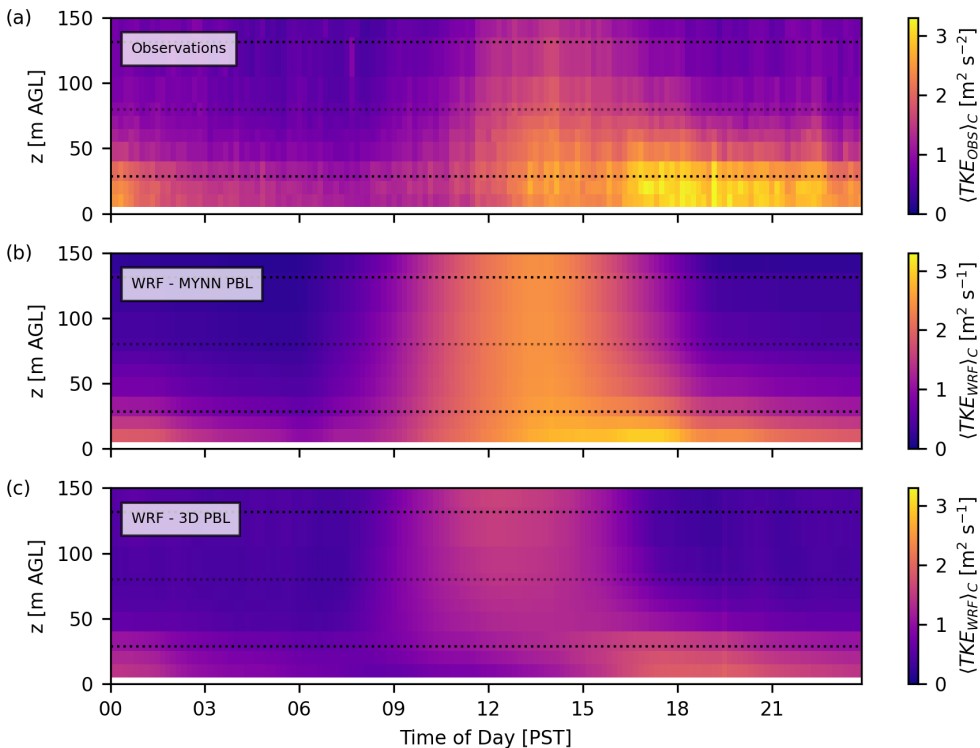

**Figure 6.** Diurnal composite-average TKE at the WOP lidar site, shown for the observations (a), the 3D PBL model configuration (b), and the MYNN model configuration (c). Dotted lines indicate the rotor-swept area of the most prevalent generic turbine model in the simulations, with hub-height $H = 80$ m and rotor diameter $D = 103$ m (NREL-1.7; see Table 1).

predictions in the midday (see Figure 4, 1200–1500 PST), although the near-surface jet-like flow is not captured accurately by the model. During and after the peak of the speedup flow (1800–0900 PST), the observations and both model configurations
show increased TKE near the surface, with reduced values aloft.

In their cold-air pool case study, Arthur et al. (2022) also found that the 3D PBL scheme with the PBL approximation predicts lower TKE values as compared to MYNN, and that times of reduced TKE values in the 3D PBL configuration were associated with improved velocity profile predictions. It is important to note that modeled TKE predictions depend on parameters such as the turbulence length scale and closure constants, which differ in the between the MYNN and 3D PBL schemes as configured
here (and in Rybchuk et al., 2022; Arthur et al., 2022; Wiersema et al., 2023). These parameters were not varied in the present study, although the reader is referred to Arthur et al. (2022) for a discussion of model sensitivity.

### 3.2 Horizontal variability

MesoWest stations are used to examine horizontal variability in model performance around the APWRA turbines. MesoWest wind data are generally available at 10 m AGL (Mesonet, 2023). Here, wind speed data are used at select stations shown

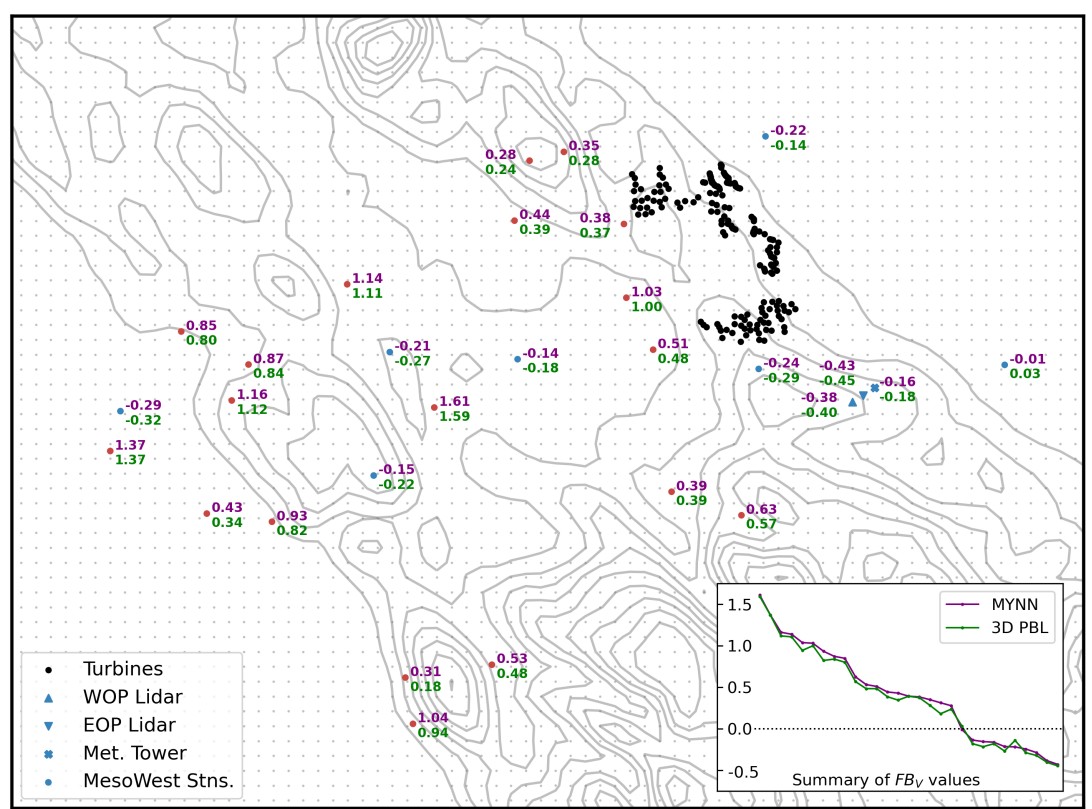

**Figure 7.** Fractional wind speed bias $FB_V$ at 10 m AGL for MYNN (purple) and 3D PBL (green) configurations at meteorological observation stations in the APWRA. Station markers are colored by the sign of the bias in the MYNN configuration, blue for negative and red for positive. Gray contour lines are shown at 100 m intervals between 100 and 1000 m AGL, and gray dots represent cell centers on the $\Delta x = 1$ km model grid. The portion of the domain shown here is highlighted by the dotted-line box in Figure 1b. Inset is a summary of 10-m $FB_V$ at all stations, sorted in descending order based on the value for the MYNN configuration.

in Figures 1 and 7. For clarity in the analysis, only stations along the primary wind direction (230-250°; see Figure 3c) are considered. Furthermore, overlapping stations and those reporting predominantly 0 m s$^{-1}$ velocity readings are excluded.

The fractional bias, defined in equation 3, is used to evaluate the spatial variability of model wind speed errors. $FB_V$ is similar to the $NMAE_V$ metric defined in equation 4, but it includes the sign of the error. While this value tends to be small over the full profile due to averaging over both positive and negative bias values at different measurement heights (see Table 2

and Figure 4), at a single height it more reliably quantifies model over- vs. underestimates.

Spatial evaluation of model performance shows that the 3D PBL scheme tends to reduce model overestimates of the 10 m wind speed relative to MYNN. As summarized in the inset of Figure 7, the 3D PBL configuration has a lower 10-m $FB_V$ value at all but 1 of the 20 stations with positive bias. At the 8 locations with negative bias, the value for the 3D PBL configuration

tends to be more negative, as is true at both lidar sites and the meteorological tower. This suggests that model underestimates
are related to near surface jet-like flows (as shown in Figure 4). However, additional vertical profile data would be necessary
for confirmation.

## 4   Wind energy predictions

### 4.1   Hub-height and rotor-equivalent wind speeds

To better establish the utility of the 3D PBL scheme for wind energy applications, model evaluation is extended to wind energy-
specific quantities, including hub-height and rotor-equivalent wind speeds. The rotor-equivalent wind speed $V_{EQ}$ is often used
in wind energy resource and turbine performance assessment (Wagner et al., 2014), and is recommended by the International
Electrotechnical Commission (IEC) for determining power curves and annual energy production (see Van Sark et al., 2019).
$V_{EQ}$ more accurately captures the kinetic energy flux through the rotor-swept area, as compared to a single hub-height wind
speed measurement $V_{HH}$. However, substantial differences between $V_{EQ}$ and $V_{HH}$ are generally only seen at times of high
shear (e.g., Van Sark et al., 2019; Redfern et al., 2019).

Following Wagner et al. (2014), the rotor-equivalent wind speed is defined as

$$V_{EQ} = \left( \sum_{i=1}^{N_h} V_i^3 \frac{A_i}{A} \right)^{1/3} \tag{7}$$

where $N_h$ is the number of observation heights, $A$ is the total rotor-swept area, and

$$A_i = \int_{z_i}^{z_{i+1}} 2\sqrt{R^2 - (z - H)^2} dz \tag{8}$$

is the area of the rotor disk segment corresponding to the $i^{th}$ observation height, with rotor radius $R$ and hub height $H$. The
integral in equation 8 is evaluated analytically with $z_i$ and $z_{i+1}$ representing the lower and upper bounds of the $i^{th}$ rotor disk
segment, which are by definition located halfway between available observation points. Here, $V_{EQ}$ is calculated using both
HilFlowS lidar data and model predictions. The modeled wind speed profiles are interpolated to the lidar observation locations
as in the bias calculations in Section 3.

As in Figures 3 and 4, a diurnal composite average captures the trend of the hub-height and rotor-equivalent wind speeds
during the study period (see Figure 8 for WOP and Figure 9 for EOP). The observed hub-height wind speed gradually increases
over the course of the day, reaching a peak around 1800 PST. It then decreases gradually, reaching a minimum around 0900
PST. The observed rotor-equivalent wind speed follows a similar trend. Note that here, $V_{EQ}$ is calculated with a hub height
$H = 80$ m and a rotor diameter $D = 103$ m, which corresponds to the most prevalent generic turbine model in the simulations
(NREL-1.7; see Table 1) and is also representative of most APWRA turbines (see discussion in Section 2.2.2 and Table 1).

The hub-height and rotor-equivalent wind speeds are generally underestimated at both sites during the ramp-up portion of
the speedup event (0900–1500 UTC). The 3D PBL configuration shows improved predictions during this time, reducing the

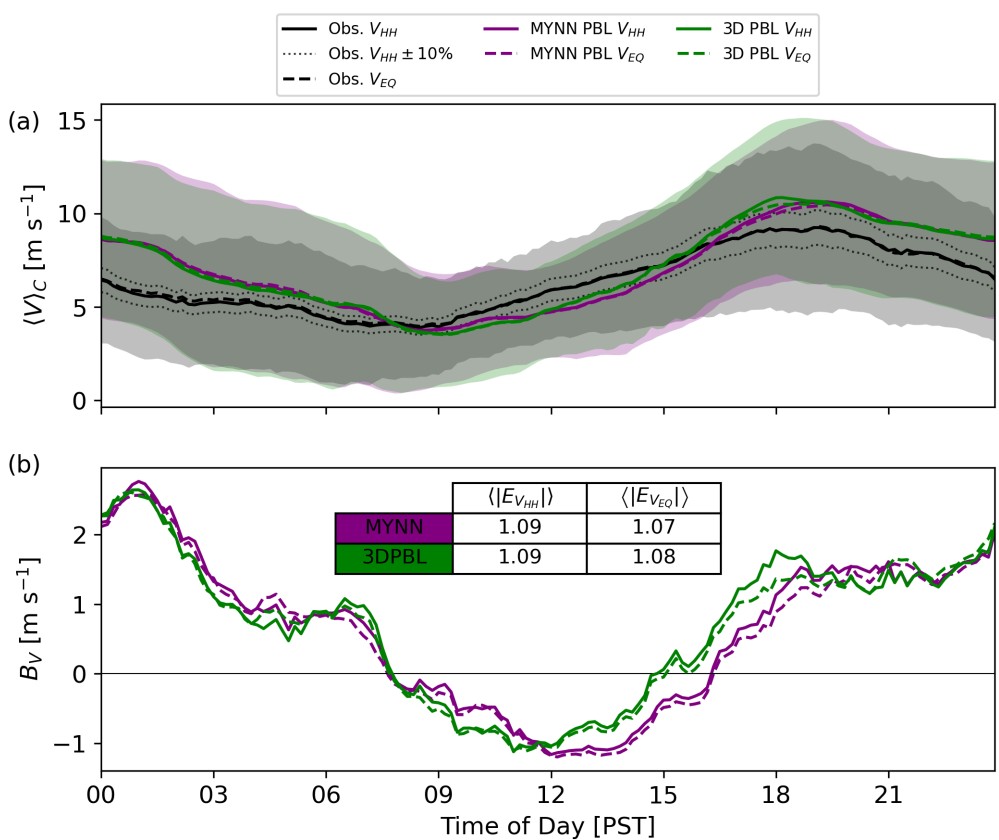

**Figure 8.** Diurnal composite-average hub-height wind speed $V_{HH}$ and rotor-equivalent wind speed $V_{EQ}$. (a) Results for WOP lidar observations, including potential mean error bounds of $\pm 10\%$ following Bingöl et al. (2009), and both model configurations; (b) model bias, including a summary of time-averaged absolute error values in m s$^{-1}$. In (a), the shaded regions show $\pm 1$ standard deviation over the diurnal composite for $V_{HH}$. $V_{EQ}$ is calculated with hub height $H = 80$ m and rotor diameter $D = 103$ m, corresponding to the most prevalent generic turbine model in the simulations (NREL-1.7; see Table 1).

negative bias by as much as 50%. Then, during the peak and decreasing portion of the speedup event, the modeled hub-height and rotor-equivalent wind speeds are generally overestimated (1500–0900 UTC), by as much as a factor of roughly 2. While

the 3D PBL configuration shows larger overpredictions than the MYNN configuration at the peak of the speedup event, its performance is similar to or slightly better than MYNN for the rest of the night. Hub-height and rotor-equivalent wind speeds for the sample day shown in Appendix A reinforce these composite-average trends.

The difference between the observed hub-height and rotor-equivalent wind speeds is larger at EOP than at WOP, highlighting differences in vertical shear between the sites despite similar wind climatology overall. As shown in Wharton and Foster (2022),

the EOP site has lower wind speeds in the bottom half of the rotor layer for an 80-m turbine, causing $V_{EQ}$ to be lower than $V_{HH}$ (see Figure 7b therein). This variability is not captured in the model, which predicts similar hub-height and rotor-equivalent

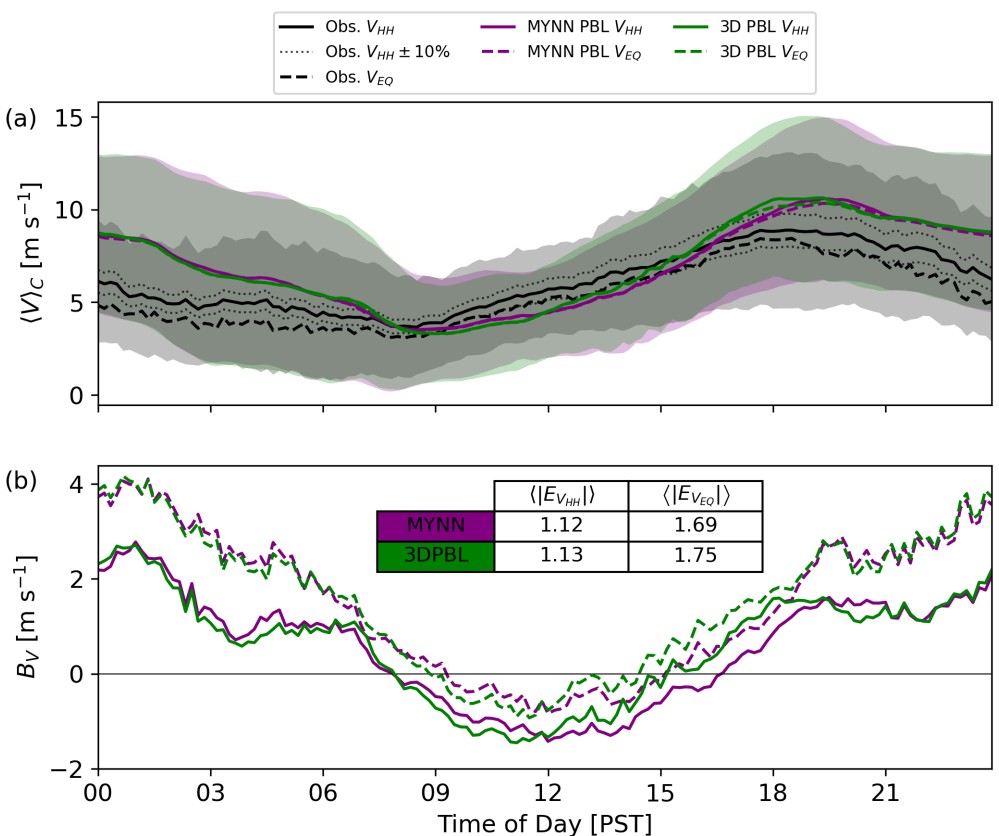

**Figure 9.** As in Figure 8, but for the EOP site.

wind speed values at both sites. Thus, at the EOP site, model bias values are larger for $V_{EQ}$, by as much as 2 m s$^{-1}$ compared to $V_{HH}$. At the WOP site, bias values for both quantities are similar. This analysis demonstrates the potential effect of using $V_{EQ}$ when evaluating model performance for wind energy applications in regions with highly sheared wind speed profiles.

### 4.2 Monthly capacity factors

Although the flows at the HilFlowS lidar locations are expected to be representative of those experienced by the APWRA turbines, more localized effects may contribute to turbine performance (see, e.g., Wharton et al., 2015; Bulaevskaya et al., 2015). For this reason, the Fitch et al. (2012) WFP is used in both model runs to represent the interaction between the APWRA turbines and the diurnal speedup events. Because Rybchuk et al. (2022) considered only an ocean environment with no terrain in their testing of the 3D PBL-WFP implementation, the present case study presents an opportunity to further evaluate the implementation in a realistic complex-terrain scenario.

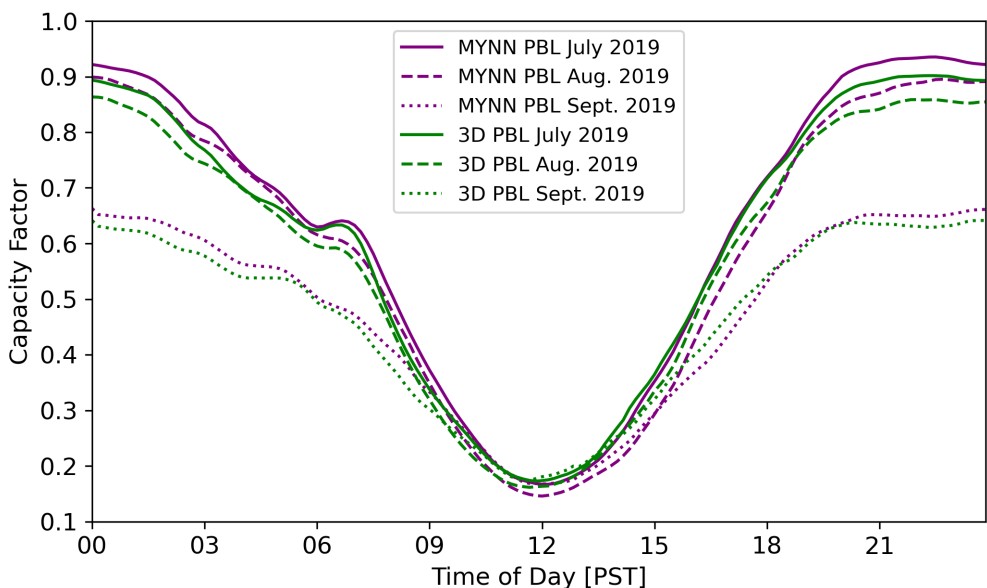

**Figure 10.** Diurnal composite-average capacity factor, by month during the study period, for modeled APWRA turbines.

Diurnal composite-average capacity factors for the WFP-modeled APWRA turbines are shown by month in Figure 10 to illustrate changes in production over the roughly 3-month-long study period. The overall trend is similar to that shown in Figure 2, with the highest capacity factors in July, a slight decrease in August, and a more substantial decrease in September. However, the same diurnal trend remains, indicating the prominence of the speedup flows throughout the mid-to-late summer.

The capacity factors in Figure 10 follow the trend of the hub-height and rotor-equivalent wind speeds at both lidar sites (shown in Figure 8 and Figure 9). Notably, however, there is a roughly 3-hour delay in the timing of the peak and minimum capacity factors relative to the modeled wind speeds at the HilFlowS lidar sites. This suggests differences in the timing of the speedup flows between the HilFlowS site and the APWRA, despite their relative proximity, and highlights the influence of terrain on power production.

To further evaluate the performance of the 3D PBL-WFP configuration during the HilFlowS study period, modeled monthly capacity factors are compared to those calculated with publicly available data (Figure 11). The EIA collects monthly plant-level generation data within the United States (EIA, 2023a, b, as shown in Figure 2). These data are depicted in Figure 11 (black bars) as an average over the five wind plants shown in Table 1, weighted by rated plant capacity. Because plant-level information is not available in WRF output, modeled monthly capacity factors in Figure 11 (colored bars) are shown as an average over the APWRA as a whole.

Overall, the modeled monthly capacity factors follow the decreasing trend evident in the EIA data. However, the model generally overestimates the reported values by roughly 7–11%. Several factors likely contribute to this overestimate. Most

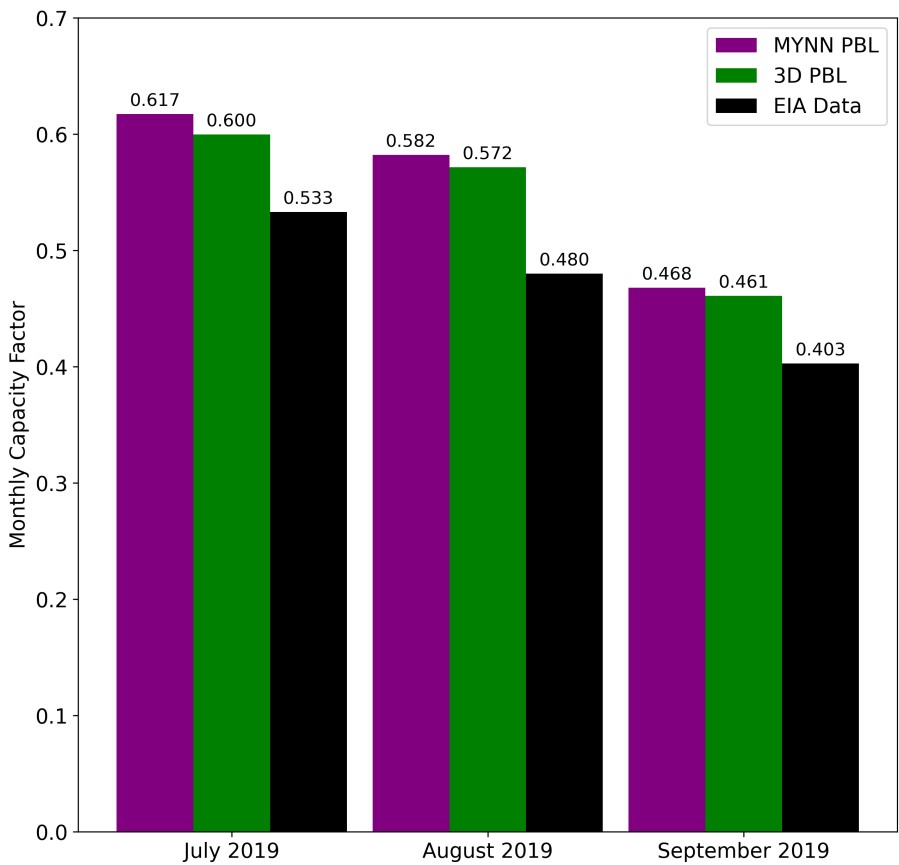

**Figure 11.** Comparison of modeled vs. EIA-reported (EIA, 2023a, b) monthly capacity factors in the APWRA during the study period.

notably for this study, overestimated wind speeds in the model, especially during the night (see Figures 3, 4, 8, and 9), likely
lead to overestimated power production. Additionally, the model does not account for turbine downtime, for example, due to
curtailment or maintenance, which reduces the reported monthly production; this likely also contributes to model overestimates.

Keeping these caveats in mind, the 3D PBL configuration predicts slightly lower monthly capacity factors relative to the
MYNN configuration (roughly 1% or less, see Figure 11). However, differences are more pronounced in the monthly diurnal
composite-average comparisons, especially at night (see Figure 10, 1800–0600 PST), when the capacity factors in the 3D PBL
configuration are up to roughly 6% smaller than those in the MYNN configuration. These results, along with those in Figures 4,
8, and 9, suggest that the 3D PBL scheme's wind power predictions may be slightly closer to reality. However, comparisons to
higher-frequency (e.g., hourly) turbine- or plant-level data are necessary for a more robust evaluation.

Although turbine- and plant-level data are not output by the WFP, grid cell-level data reveal some spatial variability in
modeled monthly capacity factor. Figure 12 shows the capacity factor and total capacity in each model grid cell that contains
turbines. Results are based on the 3D PBL configuration, although those for the MYNN configuration are qualitatively similar.

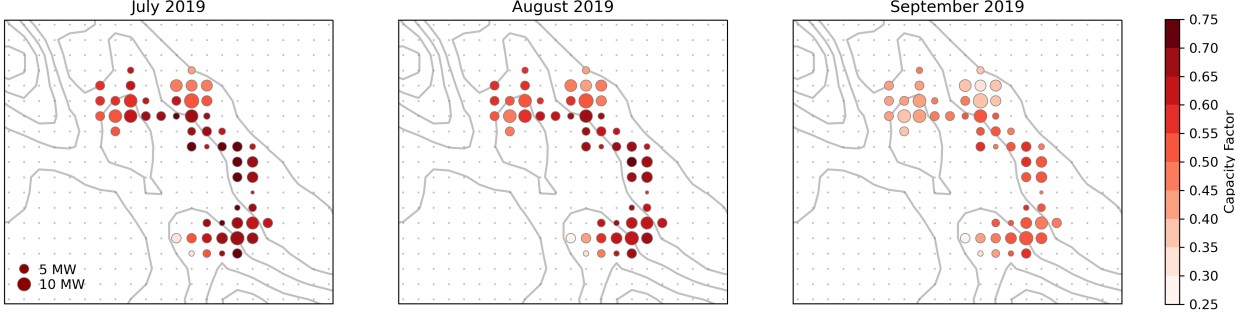

**Figure 12.** Spatial variability of modeled monthly capacity factors in the APWRA during the study period, using data from the 3D PBL configuration. Circles are shown for each model grid cell that contains turbines; the color scale represents the capacity factor and the size of the circle represents the total capacity in the given cell. Gray contour lines show the terrain at 100 m intervals between 100 and 1000 m AGL, and gray dots show cell centers on the $\Delta x = 1$ km model grid.

The capacity factor tends to be higher in the central to southeastern portion of the APWRA, where the southwesterly speedup flows are less obstructed by upstream terrain. This trend is consistent across the three months of the study period, although the overall capacity factors decrease noticeably in September. It should be noted that the Summit Wind plant, which became operational in 2021 after the study period, is located in the central APRWA to the southwest of the turbines considered here (see Hoen et al., 2018). This location is generally upstream of other plants during the summertime and likely takes advantage of the spatial trend in capacity factor seen in Figure 12. However, spatial variability in the APWRA capacity factor is expected to change seasonally due to shifts in the synoptic forcing and the predominant wind direction.

## 5    Conclusions

This study examined mesoscale model predictions of boundary layer winds and turbulence in the Altamont Pass Wind Resource Area of California, where the diurnal regional seabreeze and associated terrain-driven speedup flows drive wind energy production during the summer months. The recurring nature of these terrain-driven wind accelerations, as well as their importance to the wind energy industry, makes the APWRA a useful testbed for numerical weather prediction. In particular, this study focused on the treatment of turbulence in mesoscale models, which require a PBL scheme to parameterize subgrid-scale turbulent mixing. The WRF-based 3D PBL scheme of Juliano et al. (2022), which treats both vertical and horizontal turbulent mixing (here, using the PBL approximation), was evaluated in comparison to a traditional 1D PBL scheme, MYNN, which treats only vertical turbulent mixing.

Both PBL treatments were tested during the nearly 3-month-long HilFlowS experiment (Wharton and Foster, 2022), which took place near the APWRA in the summer of 2019. As noted by Banta et al. (2020) in their study of recurring marine-air intrusions, capturing repeated flow dynamics, and thus repeated model errors, allows for robust model evaluation. Here, as

in Banta et al. (2020), composite averaging was used to analyze model errors over the course of the study period. Model predictions were evaluated against data from two profiling lidars and a meteorological tower deployed during HilFlowS, as well as surface meteorological stations within the MesoWest network. Thus, both vertical and horizontal variability in model performance was examined.

In terms of overall model skill, the 3D PBL and MYNN configurations performed similarly over the duration of the study period, with both capturing the general timing and direction of the speedup flows but overestimating their magnitude within a typical wind turbine rotor layer. Additionally, neither model configuration captured the persistent jet-like flow observed by the lidars, and thus both models underestimated near-surface wind speeds. Similar performance between the two configurations suggests that both are limited by the chosen mesoscale resolution, which does not fully represent the effects of complex terrain on local wind profiles. It follows that in the present case study, strong synoptic conditions may drive model performance more than the PBL scheme.

Despite overall similarities in performance, several minor differences were found between PBL treatments. In terms of vertical variability, the 3D PBL scheme demonstrated slightly improved predictions of wind speed profiles during the afternoon acceleration phase of the diurnal speedup flows, and this was associated with reduced TKE relative to MYNN. Additionally, the 3D PBL scheme showed evidence of a more pronounced near-surface jet and reduced wind speeds aloft. Although this evidence is muted in the diurnal composite average, it is more pronounced on a given sample day (see Appendix A). In terms of horizontal variability, the 3D PBL scheme showed reduced positive wind speed bias at most MesoWest surface stations within the APWRA. This suggests that it more accurately captures horizontal variability over complex terrain.

In future studies, the use of increased horizontal resolution could help to further distinguish 3D PBL performance relative to MYNN. As model grid spacing progresses further into the gray zone, larger horizontal gradients will be resolved, leading to differences in flow predictions. The 3D PBL scheme has been tested successfully in the past with horizontal grid spacing between 250 and 750 m (Juliano et al., 2022; Arthur et al., 2022; Wiersema et al., 2023). Note that careful model setup, including use of the PBL approximation, is still generally required to ensure model stability. With further development of the 3D PBL scheme to improve stability, additional gains relative to MYNN or other 1D schemes may be found. Ultimately, however, accurate simulation of the observed jet-like flow at the HilFlowS site will likely require increased vertical resolution and the use of an LES closure scheme.

To further evaluate the 3D PBL scheme for wind energy applications, the mesoscale wind farm parameterization of Fitch et al. (2012) was employed. The WFP was recently coupled to the 3D PBL scheme by Rybchuk et al. (2022) and tested in an idealized ocean environment. The present study provided an opportunity to test the 3D PBL-WFP implementation, as compared to the standard WRF implementation with MYNN, in a realistic complex-terrain scenario. Overall, the 3D PBL-WFP performs similarly to the MYNN-WFP, providing additional confidence in the implementation.

Modeled capacity factors capture the general diurnal trend of the observed speedup flows, but are roughly 7–11% larger than EIA-reported values in the APWRA. This is likely due to overestimated wind speeds during the peak and decelerating phase of the speedup events, as well as other factors including turbine operation and differences between the modeled and actual turbines. However, because wind power is proportional to the cube of wind speed over much of a turbine's operational

range, small relative improvements in the modeled wind speed translate to more noticeable improvements in modeled power production. Consistently over the 3-month study period, the 3D PBL configuration reduced overestimates of monthly capacity factors, relative to the MYNN configuration.

In closing, this study has helped to establish the utility of the 3D PBL scheme for wind energy applications in complex terrain. Its overall similar performance to MYNN, a much more established PBL scheme, is encouraging, as is evidence of improved performance under certain conditions and across the spatially heterogeneous APWRA. However, the 3D PBL scheme requires additional development and testing to confirm its robustness. As mentioned above, the 3D PBL scheme allows more run-time flexibility in turbulence treatment relative to MYNN and other 1D PBL schemes, which could facilitate rapid performance improvements. Ultimately, increased understanding of model sensitivity to grid spacing and turbulence closure parameters (e.g., length scales, closure constants, and use of the PBL approximation) will guide the use of the 3D PBL scheme for high-resolution numerical weather prediction and wind energy applications.

*Code and data availability.* All HilFlowS observational data used in this work are publicly available through the United States Department of Energy's Atmosphere to Electrons Data Archive and Portal (https://a2e.energy.gov/about/dap); each dataset is cited individually in the main text. MesoWest data are available through Mesonet (2023). The WRF code used in this work is available on GitHub at https://github.com/twjuliano/WRF/tree/develop_3dpbl_on_top, commit f04c02387bdf9f3ab5f93a1b4b28c5f35c05a950. The WRF configuration files are available on Zenodo (see Arthur, 2024). Modeled wind turbine specifications are based on data from NREL (2022) and wind-turbine-models.com (2024b, a), as described in the text and summarized in Table 1.

**Appendix A: Sample day**

To complement the composite-average wind speed results shown in the main text, this appendix shows results from a sample day during the study period: 21 July 2019. This day was chosen to highlight differences between the 3D PBL and MYNN configurations while also showing consistency with the composite-average results. The same day was highlighted in the original HilFlowS study (Wharton and Foster, 2022, see Figure 5 therein). Wind speed profiles at WOP are shown in Figure A1, corresponding to Figure 4 in the main text. Hub-height and rotor-equivalent wind speed time series at WOP are shown in Figure A2, corresponding to Figure 8 in the main text.

On this day, during the peak of the evening speedup flow (1800–0300 PST), the 3D PBL configuration predicts a more pronounced jet-like wind speed profile (with its wind speed maximum closer to the surface) than the MYNN configuration. This leads to improved predictions of the hub-height and rotor-equivalent wind speeds. However, both model configurations overestimate the observed rotor-layer wind speed during this time, while underestimating the near-surface wind speed. There is also evidence of reduced error for the 3D PBL configuration during the onset of the speedup event (0900–1500 PST). These results are generally consistent with the composite average, while also highlighting potential model improvements when the 3D PBL configuration is used.

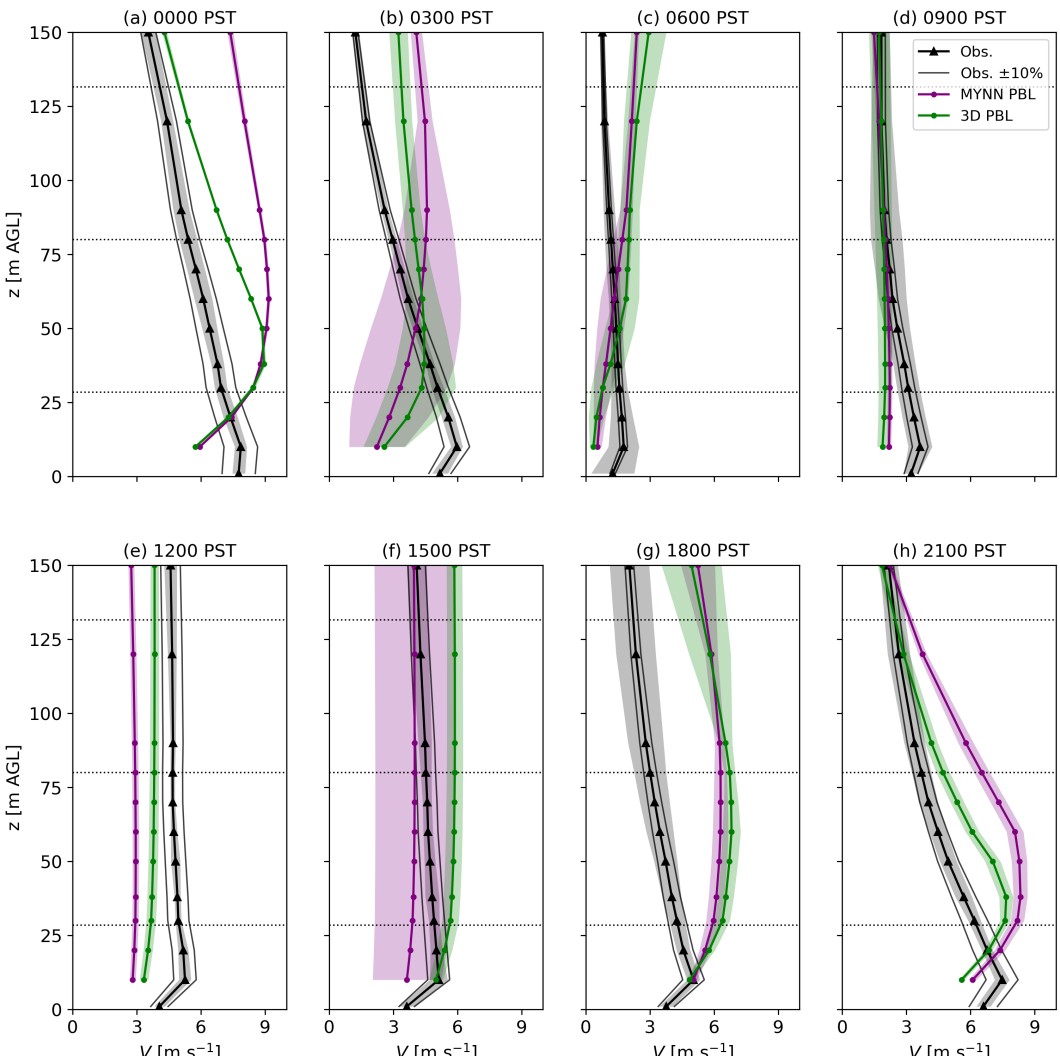

**Figure A1.** Wind speed profiles on 21 July 2019, shown for WOP lidar observations and both model configurations. Potential mean error bounds of $\pm 10\%$ are also shown for the lidar observations following Bingöl et al. (2009). Profiles are averaged over the hour indicated at the top of each panel, and model data have been interpolated to the vertical levels of the lidar. Note that data is included from the lidar's on-board meteorological station at 1 m AGL, but model errors are not evaluated at this height. The shaded regions show $\pm 1$ standard deviation over the given hour of the sample day. Dotted lines indicate the rotor-swept area of the most prevalent generic turbine model in the simulations, with hub-height $H = 80$ m and rotor diameter $D = 103$ m (NREL-1.7; see Table 1).

*Author contributions.* Writing–original draft preparation, formal analysis, and visualization: RSA; writing–review and editing: AR, TWJ, GR, SW, JKL, JDF; software, AR, TWJ, JKL: data curation: SW, GR; conceptualization: RSA, GR, SW, JDF; funding acquisition and project administration: JDF, RSA.

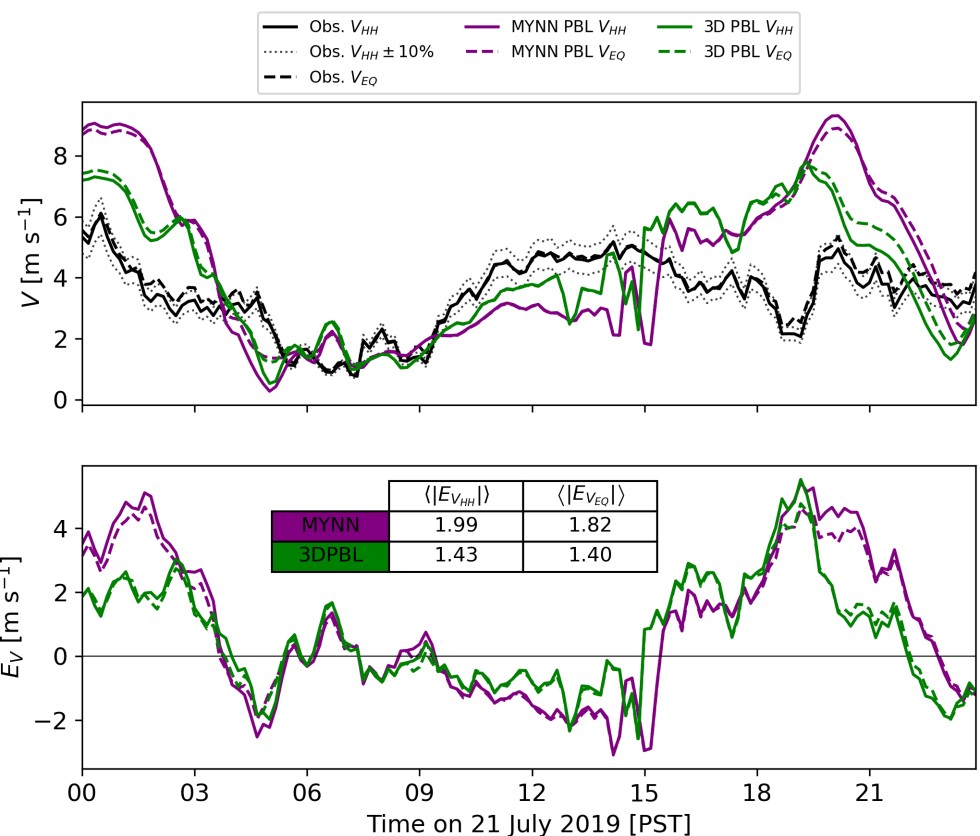

**Figure A2.** Hub-height wind speed $V_{HH}$ and rotor-equivalent wind speed $V_{EQ}$ on 21 July 2019. (a) Results for WOP lidar observations, including potential error bounds of $\pm 10\%$ following Bingöl et al. (2009), and both model configurations; (b) model error, including a summary of absolute error values (in m s$^{-1}$) time-averaged over the day. $V_{EQ}$ is calculated with hub height $H = 80$ m and rotor diameter $D = 103$ m, corresponding to the most prevalent generic turbine model in the simulations (NREL-1.7; see Table 1).

*Competing interests.* The authors declare that they have no conflict of interest.

*Acknowledgements.* This research was supported by the U.S. Department of Energy (DOE), Office of Energy Efficiency and Renewable Energy, Wind Energy Technologies Office. This work was performed under the auspices of the U.S. Department of Energy by Lawrence Livermore National Laboratory under Contract DE AC52-07NA27344. Pacific Northwest National Laboratory is operated by Battelle Memorial Institute for the U.S. Department of Energy under Contract DE-AC05-76RL01830. This work was authored in part by the National Renewable Energy Laboratory, operated by Alliance for Sustainable Energy, LLC, for the U.S. Department of Energy under Contract No.

DE-AC36-08GO28308. The views expressed in the article do not necessarily represent the views of the DOE or the U.S. Government. The U.S. Government retains and the publisher, by accepting the article for publication, acknowledges that the U.S. Government retains a

nonexclusive, paid-up, irrevocable, worldwide license to publish or reproduce the published form of this work, or allow others to do so, for U.S. Government purposes. Co-author TWJ is grateful for support in part from the Department of Energy Wind Energy Technologies Office through Contract No. DE-AC05-76RL01830 to Pacific Northwest National Laboratory (PNNL). The NSF National Center for Atmospheric Research (NSF NCAR) is a subcontractor to PNNL under Contract No. 659135. NSF NCAR is a major facility sponsored by the National Science Foundation under Cooperative Agreement No. 1852977. The authors acknowledge Chris Golaz and Tom Edmunds for their assistance with the EIA data, Tianyi Li for his assistance in calculating rotor-equivalent wind speeds, and two anonymous reviewers for their suggestions to improve the manuscript.

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
