# Peer review of "Evaluating mesoscale model predictions of diurnal speedup events in the Altamont Pass Wind Resource Area of California"

_Wind Energy Science, 2024_

## Referee Comment (RC1)

Arthur et al
**Evaluating mesoscale model predictions of diurnal speedup events in the Altamont Pass Wind Resource Area of California**
Manuscript wes-2024-137

Introduction

This manuscript presents a comparison of season-long (~75 days) simulations of atmospheric flow at a wind power production site located in hilly terrain in central California. Simulations are performed with the WRF model at 1-km horizontal grid spacing and with two alternative parameterization schemes for planetary-boundary layer turbulence. The first PBL parameterization is the well-known and widely adopted MYNN scheme, which models only vertical mixing and additionally relies on Smagorinsky-type horizontal diffusion. The second PBL parameterization is a recently developed three-dimensional scheme that represents horizontal mixing in a physically consistent manner. Given the complex-terrain nature of the site, it is expected that horizontal heterogeneity of the turbulence could be misrepresented in a purely one-dimensional parameterization. Therefore, it is expected that the 3D PBL scheme has greater forecast skill.
This wind farm features prominent diurnal variability of the wind field, with speed-ups in the late afternoon/evening driven by differential heating of the atmosphere. Average diurnal cycles of the vertical profiles of wind speed, wind direction, temperature, vertical velocity, turbulent kinetic energy at three measurement sites are derived from the simulations and compared with observations from two wind lidars and a meteorological tower. The analysis deals in depth with data from one lidar site and from the tower. The primary purpose of the analysis is to evaluate if the new 3D PBL parameterization compares more favourably with observations. A secondary purpose is to verify that coupling the new 3D PBL scheme with an existing wind farm parameterization provides reasonable results.

Recommendation

The manuscript is well written and contains clear and well-designed figures. The research is clearly relevant to the scope of the journal. The focus on boundary-layer modelling over complex terrain makes this work interesting in a broader context beyond wind energy, e.g., for mountain meteorologists. In fact, the introductory chapter is an excellent concise crash course on the challenges of boundary-layer modelling over mountains.
The analysis is generally solid. The only aspect I find questionable is a degree of confusion between the systematic and random components of model error (see comments 1-2-3 below). The results are probably less compelling than one might expect, in that the impact of the new turbulence parameterization is rather weak (see comments 4-5). In fact, the most relevant systematic errors (persistent wind speed underestimation near the ground, and overestimation across the rotor diameter during the evening speed-up) are corrected only marginally by the new scheme. However, the small difference in the skill of wind forecasts translates into a somewhat more marked improvement of the power production.
The manuscript does not break new scientific ground, but the results will likely be of interest for those who use WRF simulations for wind resource forecasting and assessment. Acceptance is recommended, conditional to minor revisions.

General comments

1. Equation 1 defines the "model bias". An instantaneous, local deviation between model and observation is NOT bias. At a given time and place, model error has both a systematic and a random component. Strictly speaking, bias is the *systematic* component of the deviation between model and observations. Thus, it is the *average* difference evaluated over a sample. I would recommend adding an averaging operator to the RHS of the equation. Averaging can consider multiple dimensions (e.g. time only, or time and height), and therefore I would recommend that the authors accurately define the averaging operations.

2. Line 278-279. Here I see another example of confusion between systematic and random error. "The expected maximum error is smaller than the standard deviation of the diurnal composite". From the preceding text (around line 115), I understood that the estimate by Bingöl et al refers to the systematic (mean) component of the model error (accuracy). In this figure, it seems to be used as a standard deviation (precision). I'm not sure this is appopriate, because the standard deviation represents deviations from the mean; thus the random component of the model error, not the systematic one.
   A side note: There are two shades of gray around the observed profiles in Figure 4. The potential 10% error in the observations is likely the narrowest shaded area (plus-minus 1 m/s). The broadest one (plus-minus 3 m/s) likely represents the standard deviation. Please clarify.

3. Figure 11 and line 392. Here I see yet another example of misinterpretation of model error statistics. The mean absolute error of the wind speed is regarded as a measure of bias. It is not! The mean absolute error conveys essentially the same information as the root mean square error, but using a different norm: the absolute value norm instead of the Euclidean norm. A sensible measure of bias would be the mean error.
   Furthermore, I'm not sure Fig. 11 provides useful information. The authors argue that differences in "bias" among the simulations with the two schemes explain errors in the modelled capacity factors. I would buy the argument if the dots (one dot every 10 minutes in a full diurnal cycle) were well aligned on the diagonal. However, quadrants I and III together contain roughly the same number of dots as quadrants II and IV together.
   All considered, I would recommend removing this figure.

4. A general remark on Fig. 4, 5, 7, 8: the differences between MYNN and 3D PBL simulations are always very small in comparison to the respective deviation from the observations. This applies to the mean wind speed profiles (Fig. 4), temperature profiles (Fig. 5), areal distribution of the wind speed bias (Fig. 7), diurnal cycle of the wind speed bias (Fig. 8). This fact could be interpreted in two ways: either (i) the PBL scheme is a minor contributor to total model error; or (ii) neither of the two PBL schemes is able to successfully reduce model error. None of these conclusions is particularly encouraging. Can the authors comment on that?

5. The conclusions (line 450-454) state that "several notable differences were found between PBL treatments" and that "the 3D PBL scheme showed evidence of a more pronounced near-surface jet and reduced wind speeds aloft". There are differences, indeed, but honestly they seem rather minor (see comment 4). It would be fair acknowledge this fact, and to discuss openly the possible reasons for the limited impact of the new PBL scheme. Also, it could be useful to point out that, because wind power is proportional to the cube of wind speed, small relative improvements in modelled wind speed translate in noticeable improvements in modelled power production (or modelled capacity factor).

Specific comments

6. Line 119: I guess the "dynamic conversion factors" are a way to quantify systematic errors in wind speed simulations. Could the authors clarify, and maybe use one or two sentences to explain how these corrections are computed?

7. Lines 158-159: My understanding of the boundary-layer approximation, which usually applies to horizontally homogeneous boundary layers, is that all horizontal gradients of mean quantities (and the turbulent fluxes that are assumed to be proportional to them) are neglected. You probably mean something more subtle here. To help the reader, could you please spend a few more words to clarify?

8. Lines 168-169: "positive-definite 6th order diffusion". It would be good to specify that this (pseudo-)horizontal diffusion is computational, not physical. I guess it is used to help dissipate 2dx noise and maintain numerical stability.

9. Line 215 explains that bias is computed also for the wind direction. It is a delicate operation, because of the cyclic nature of the variable. Consider the case of an observed timeseries of wind directions such as 0, 1, 359; and a modelled timeseries such as 358, 359, 1. The

model is actually very accurate (the error is 2 degrees at most), but computing the mean error without accounting for the periodicity yields values around 180 degrees. Could the authors explain how they circumvent this problem?

10. Figure 3: In looking at this figure I wondered if the slopes of the modelled and true terrain are similar or not. The text says (line 88) that the WOP site is on a ridgeline, but the modelled orography in Fig. 1 looks like an eastward facing slope. Low-level wind vectors are subject to a parallel flow condition, and if the modelled slope geometry is somewhat inaccurate, biases in the u and w components will inevitably follow. This aspect might be worth commenting.

    Furthermore, the authors state that "while the model captures some negative vertical velocities at the study site during the speedup events, its vertical velocities are too weak and thus do not translate to near-surface accelerations of the magnitude seen in the observations." I understand that this reasoning is based on incompressible mass continuity arguments. However, the vertical convergence/horizontal divergence concept is seemigly only relevant to the evening and night hours. Fig. 3b instead shows that the horizontal wind speed at low levels is  underestimated also at daytime (when vertical velocity is predominantly upward). Is there a different explanation for the daytime bias?

11. Line 255: "The vertical velocity error is not normalized because w has both positive and negative values". I'm not sure I understand. You mean that, by normalizing, you would likely run into divisions by zero in Eq 2 and 3?

Technical corrections

12. Line 87: "Diablo Range". This geographical name is unexplained, and is unlikely to be broadly known. Please label the site in Figure 1, or describe its position in the figure caption.

13. Figure 1: It may be a pet peeve of mine, but I'm allergic to terrain colormaps with deep blue shades in inland areas. It looks like the figures were plotted with python/matplotlib. If so, it is fairly straightforward to truncate colormaps; e.g. using only the upper 80% of the range (green and brown shades).

14. Figure 2, caption: "Capacity factors", first mentioned here. It might be worth to explain the term. It is a basic concept, but casual readers might not know it.

15. Line 128: Reference to "(Synoptic, 2023)". This is likely a broken reference. See also line 323.

16. Table 1: Symbols (Mfr-PR, H, D) have rather obvious meanings, but it would be good to explain them in the caption.

---

## Author Comment (AC1)

We thank the referees for their careful reading of our manuscript and for their suggestions. Please find our responses below in blue. Note that line numbers refer to the main pdf, not the version with changes tracked.

**Response to RC1**

**Introduction**

This manuscript presents a comparison of season-long (~75 days) simulations of atmospheric flow at a wind power production site located in hilly terrain in central California. Simulations are performed with the WRF model at 1-km horizontal grid spacing and with two alternative parameterization schemes for planetary-boundary layer turbulence. The first PBL parameterization is the well-known and widely adopted MYNN scheme, which models only vertical mixing and additionally relies on Smagorinsky-type horizontal diffusion. The second PBL parameterization is a recently developed three-dimensional scheme that represents horizontal mixing in a physically consistent manner. Given the complex-terrain nature of the site, it is expected that horizontal heterogeneity of the turbulence could be misrepresented in a purely one-dimensional parameterization. Therefore, it is expected that the 3D PBL scheme has greater forecast skill. This wind farm features prominent diurnal variability of the wind field, with speed-ups in the late afternoon/evening driven by differential heating of the atmosphere. Average diurnal cycles of the vertical profiles of wind speed, wind direction, temperature, vertical velocity, turbulent kinetic energy at three measurement sites are derived from the simulations and compared with observations from two wind lidars and a meteorological tower. The analysis deals in depth with data from one lidar site and from the tower. The primary purpose of the analysis is to evaluate if the new 3D PBL parameterization compares more favourably with observations. A secondary purpose is to verify that coupling the new 3D PBL scheme with an existing wind farm parameterization provides reasonable results.

**Recommendation**

The manuscript is well written and contains clear and well-designed figures. The research is clearly relevant to the scope of the journal. The focus on boundary-layer modelling over complex terrain makes this work interesting in a broader context beyond wind energy, e.g., for mountain meteorologists. In fact, the introductory chapter is an excellent concise crash course on the challenges of boundary-layer modelling over mountains. The analysis is generally solid. The only aspect I find questionable is a degree of confusion between the systematic and random components of model error (see comments 1-2-3 below). The results are probably less compelling than one might expect, in that the impact of the new turbulence parameterization is rather weak (see comments 4-5). In fact, the most relevant systematic errors (persistent wind speed underestimation near the ground, and overestimation across the rotor diameter during the evening speed-up) are corrected only marginally by the new scheme. However, the small difference in the skill of wind forecasts translates into a somewhat more marked improvement of the power production. The manuscript does not break new scientific ground, but the results will likely be of interest for those who use WRF simulations for wind resource forecasting and assessment. Acceptance is recommended, conditional to minor revisions.

We appreciate this recommendation and will address your comments in more detail below.

General comments

1. Equation 1 defines the "model bias". An instantaneous, local deviation between model and observation is NOT bias. At a given time and place, model error has both a systematic and a random component. Strictly speaking, bias is the *systematic* component of the deviation between model and observations. Thus, it is the *average* difference evaluated over a sample. I would recommend adding an averaging operator to the RHS of the equation. Averaging can consider multiple dimensions (e.g. time only, or time and height), and therefore I would recommend that the authors accurately define the averaging operations.

Thank you for this suggestion to clarify our analysis of model errors. We now define the instantaneous deviations between the model and observations in equation 1 more generally as errors. The diurnal composite bias is then defined in equation 2, including a diurnal composite time averaging operator (angle brackets with subscript $C$) on the RHS. The definitions of the time-height average error metrics (now equations 3-5) have been updated accordingly, with angle brackets used for time averages over the study period and an overbar for vertical averages over the measurement heights. Figures 3, 4, 5, 6, 8, and 9 have also been updated to include the new averaging notation.

2. Line 278-279. Here I see another example of confusion between systematic and random error. "The expected maximum error is smaller than the standard deviation of the diurnal composite". From the preceding text (around line 115), I understood that the estimate by Bingöl et al refers to the systematic (mean) component of the model error (accuracy). In this figure, it seems to be used as a standard deviation (precision). I'm not sure this is appropriate, because the standard deviation represents deviations from the mean; thus the random component of the model error, not the systematic one. A side note: There are two shades of gray around the observed profiles in Figure 4. The potential 10% error in the observations is likely the narrowest shaded area (plus-minus 1 m/s). The broadest one (plus-minus 3 m/s) likely represents the standard deviation. Please clarify.

We apologize for this confusion. On lines 316-319, we have edited the text to focus on the comparison of the potential mean lidar errors to the (mean) model bias, rather than the standard deviation:

"As a conservative estimate, the findings of Bingöl et al. (2009) imply mean horizontal wind speed errors as large as roughly 1.5 m s$^{-1}$ in the HilFlowS lidar observations (see gray bounding lines in Figure 4). In general, the expected maximum lidar error is smaller than the model bias, especially near the surface. Thus, the potential lidar error is not expected to affect the present conclusions related to model evaluation."

We have also modified Figure 4, as well as Figures 8, 9, A1, and A2, and their corresponding captions, to better distinguish the mean lidar error (now shown with bounding lines) from the standard deviation (still shown with colored shading).

3. Figure 11 and line 392. Here I see yet another example of misinterpretation of model error statistics. The mean absolute error of the wind speed is regarded as a measure of bias. It is not! The mean absolute error conveys essentially the same information as the root mean square error, but using a different norm: the absolute value norm instead of the Euclidean norm. A sensible measure of bias would be the mean error. Furthermore, I'm not sure Fig. 11 provides useful information. The authors argue that differences in "bias" among the simulations with the two schemes explain errors in the modelled capacity factors. I would buy the argument if the dots (one dot every 10 minutes in a full diurnal cycle) were well aligned on the diagonal. However, quadrants I and III together contain roughly the same number of dots as quadrants II and IV together. All considered, I would recommend removing this figure.

We agree with the referee's assessment and have removed this figure from the manuscript, along with the related discussion.

4. A general remark on Fig. 4, 5, 7, 8: the differences between MYNN and 3D PBL simulations are always very small in comparison to the respective deviation from the observations. This applies to the mean wind speed profiles (Fig. 4), temperature profiles (Fig. 5), areal distribution of the wind speed bias (Fig. 7), diurnal cycle of the wind speed bias (Fig. 8). This fact could be interpreted in two ways: either (i) the PBL scheme is a minor contributor to total model error; or (ii) neither of the two PBL schemes is able to successfully reduce model error. None of these conclusions is particularly encouraging. Can the authors comment on that?

We have commented on this in the conclusion section (see lines 475-478):

"Similar performance between the two configurations suggests that both are limited by the chosen mesoscale resolution, which does not fully represent the effects of complex terrain on local wind profiles. It follows that in the present case study, strong synoptic conditions may drive model performance more than the PBL scheme."

5. The conclusions (line 450-454) state that "several notable differences were found between PBL treatments" and that "the 3D PBL scheme showed evidence of a more pronounced near-surface jet and reduced wind speeds aloft". There are differences, indeed, but honestly they seem rather minor (see comment 4). It would be fair acknowledge this fact, and to discuss openly the possible reasons for the limited impact of the new PBL scheme. Also, it could be useful to point out that, because wind power is proportional to the cube of wind speed, small relative improvements in modelled wind speed translate in noticeable improvements in modelled power production (or modelled capacity factor).

We agree that the overall differences are relatively minor, and we have tried to soften the wording throughout the paper, especially in the conclusion section, to reflect this. In addition to the text quoted in response to #4 above, we have also expanded on potential ways to better differentiate the two PBL schemes (lines 486-493):

"In future studies, the use of increased horizontal resolution could help to further distinguish 3D PBL performance relative to MYNN. As model grid spacing progresses further into the gray zone, larger horizontal gradients will be resolved, leading to differences in flow predictions. The

3D PBL scheme has been tested successfully in the past with horizontal grid spacing between 250 and 750 m (Juliano et al., 2022; Arthur et al., 2022; Wiersema et al., 2023). Note that careful model setup, including use of the PBL approximation, is still generally required to ensure model stability. With further development of the 3D PBL scheme to improve stability, additional gains relative to MYNN or other 1D schemes may be found. Ultimately, however, accurate simulation of the observed jet-like flow at the HilFlowS site will likely require increased vertical resolution and the use of an LES closure scheme."

We also hope that the addition of the sample day in Appendix A provides more concrete evidence of differences between the PBL configurations. Please see our response to RC2 Major comment #1 below for more discussion of the new appendix.

Lastly, thank you for the suggestion related to wind power results. We have added a note to this effect on lines 502-505:

"…because wind power is proportional to the cube of wind speed over much of a turbine's operational range, small relative improvements in the modeled wind speed translate to more noticeable improvements in modeled power production. Consistently over the 3-month study period, the 3D PBL configuration reduced overestimates of monthly capacity factors, relative to the MYNN configuration."

Specific comments

6. Line 119: I guess the "dynamic conversion factors" are a way to quantify systematic errors in wind speed simulations. Could the authors clarify, and maybe use one or two sentences to explain how these corrections are computed?

That is correct. We have reworded this paragraph to better explain the conversion factors (see lines 123-126):

"An earlier experiment in the APWRA (Wharton et al., 2015) that used identical ZephIR300 lidars to measure hill speedup flows and their effects on power production assessed terrain-induced measurement errors with the Dynamics software package provided by ZephIR Ltd. As discussed therein, the software converts raw lidar line-of-sight velocity data into unbiased measurements of wind speed and wind direction for hilly sites, based on the work of Bingöl et al. (2009)."

7. Lines 158-159: My understanding of the boundary-layer approximation, which usually applies to horizontally homogeneous boundary layers, is that all horizontal gradients of mean quantities (and the turbulent fluxes that are assumed to be proportional to them) are neglected. You probably mean something more subtle here. To help the reader, could you please spend a few more words to clarify?

Yes, in response to RC2 Major comment #1 below, we have expanded our introduction of the boundary-layer approximation (now referred to as the "PBL approximation" for consistency with Juliano et al., 2022) as it applies to the 3D PBL scheme.

8. Lines 168-169: "positive-definite 6th order diffusion". It would be good to specify that this (pseudo-)horizontal diffusion is computational, not physical. I guess it is used to help dissipate 2dx noise and maintain numerical stability.

The referee is correct, we have added this clarification to the text on lines 186-191:

"…following Arthur et al. (2022), WRF's option to add positive-definite 6th-order horizontal diffusion (diff_6th_opt=2) is used in both configurations with a factor of 0.25. The added diffusion is purely numerical and is used to damp grid-scale noise. However, to prevent over-diffusion in regions of sloping terrain, where numerical diffusion is already expected to be relatively large, the added 6th-order diffusion is linearly damped between slopes of 0 and 0.05 (2.86º) and turned off for larger slopes (using the namelist options diff_6th_slopeopt=1 and diff_6th_thresh=0.05)."

9. Line 215 explains that bias is computed also for the wind direction. It is a delicate operation, because of the cyclic nature of the variable. Consider the case of an observed timeseries of wind directions such as 0, 1, 359; and a modelled timeseries such as 358, 359, 1. The model is actually very accurate (the error is 2 degrees at most), but computing the mean error without accounting for the periodicity yields values around 180 degrees. Could the authors explain how they circumvent this problem?

Thank you for this suggestion. We now explain how the instantaneous wind direction error ($E_\phi$) values are adjusted on lines 240-241:

"Note that $E_\phi$ is adjusted to account for the cyclical nature of the wind direction: if the raw $E_\phi$ value is less than -180º (greater than 180º), it is adjusted by +360º (-360º)."

10. Figure 3: In looking at this figure I wondered if the slopes of the modelled and true terrain are similar or not. The text says (line 88) that the WOP site is on a ridgeline, but the modelled orography in Fig. 1 looks like an eastward facing slope. Low-level wind vectors are subject to a parallel flow condition, and if the modelled slope geometry is somewhat inaccurate, biases in the u and w components will inevitably follow. This aspect might be worth commenting. Furthermore, the authors state that "while the model captures some negative vertical velocities at the study site during the speedup events, its vertical velocities are too weak and thus do not translate to near-surface accelerations of the magnitude seen in the observations." I understand that this reasoning is based on incompressible mass continuity arguments. However, the vertical convergence/horizontal divergence concept is seemingly only relevant to the evening and night hours. Fig. 3b instead shows that the horizontal wind speed at low levels is underestimated also at daytime (when vertical velocity is predominantly upward). Is there a different explanation for the daytime bias?

Regarding the resolved terrain, we have noted this important point more clearly on lines 304-306:

"As mentioned previously, the 1 km horizontal grid spacing of the present simulations limits the ability of the model to capture the observed jet-like flow near the surface. In particular, the hilly topography of the HilFlowS site, including the individual ridgelines on which the lidars were deployed, is not fully captured (see Figure 1)."

Additionally, we thank the reviewer for suggesting additional discussion of the daytime flows. We have added the following potential explanation to lines 287-293:

"During the onset of the speedup events, the 3D PBL configuration predicts faster wind speeds than the MYNN configuration throughout the lidar range, showing reduced negative bias compared to the observations, especially below hub height (assumed to be 80 m; Figure 4, 1200-1500 PST). This may be due to slightly improved predictions of vertical mixing of higher momentum from aloft; during this time, prior to jet development, the winds follow a standard quasi-logarithmic profile. The 3D PBL scheme has been shown previously, in idealized tests, to improve model performance during daytime convective conditions (Juliano et al., 2022)."

11. Line 255: "The vertical velocity error is not normalized because w has both positive and negative values". I'm not sure I understand. You mean that, by normalizing, you would likely run into divisions by zero in Eq 2 and 3?

We apologize for the confusion. We have removed this statement and instead simply present the $FB_w$ and $NMAE_w$ results in Table 2.

Technical corrections

12. Line 87: "Diablo Range". This geographical name is unexplained, and is unlikely to be broadly known. Please label the site in Figure 1, or describe its position in the figure caption.

We have added new description of the regional geography on lines 76-78:

"The Altamont Pass Wind Resource Area (APWRA) is a collection of wind plants located in a gap within the Diablo Range of north-central California. The gap is just east of San Francisco Bay and south of the San Francisco Bay Delta, and is roughly bounded by Mt. Diablo to the northwest and the greater Diablo Range to the southeast (see Figure 1)."

Corresponding labels have also been added to Figure 1.

13. Figure 1: It may be a pet peeve of mine, but I'm allergic to terrain colormaps with deep blue shades in inland areas. It looks like the figures were plotted with python/matplotlib. If so, it is fairly straightforward to truncate colormaps; e.g. using only the upper 80% of the range (green and brown shades).

We have updated the colormap as suggested – blue regions now explicitly denote water (as represented in the model).

14. Figure 2, caption: "Capacity factors", first mentioned here. It might be worth to explain the term. It is a basic concept, but casual readers might not know it.

We have defined capacity factors in the text where Figure 2 is first referenced (lines 82-83).

15. Line 128: Reference to "(Synoptic, 2023)". This is likely a broken reference. See also line 323.

This is a reference to the MesoWest data, accessed via the "Synoptic Data API". We have changed the citation to say (Mesonet, 2023), which we hope is more informative.

16. Table 1: Symbols (Mfr-PR, H, D) have rather obvious meanings, but it would be good to explain them in the caption.

These symbols are now defined in the caption.

**Response to RC2**

General considerations

In this contribution, the authors present three-month long simulations of wind data in the vicinity of a wind park in central California. They use WRF with two different boundary layer parameterizations, the well-known (1-dimensional) MYNN scheme and a recently introduced 3D PBL parameterization. To estimate wind power -related parameters they employ a 'wind farm parameterization' (WFP). Atmospheric data for verification stem from 2 wind lidars at some distance from the wind farm and a number of meso-net station distributed in the domain.

The goal of the study is (1) to 'evaluate the 3D PBL scheme in complex terrain' and (2) 'to test the WFP coupled to the 3D PBL scheme in a realistic configuration with terrain' (l. 68ff).

The results to support (1) are presented as average daily time-height cross sections (Figs. 3&6) or average profiles for different times (Figs. 4&5) - and in concert with the chosen error metrics do not strongly support the goal of model evaluation. The resulting fractional biases (Tab. 2) for wind speed, for example suggest an almost perfect simulation (a fractional bias of a few permille (!), what simply suggests that biases are approximately normally distributed (in space and time).

The results to support goal (2) are again presented as some average statistics and figures – which might be more informative for the wind power community (and hence the audience of the present journal).

I have got a number of major comments, which I feel need to be addressed before the paper can be recommended for publication. In addition, a number of minor comments are given at the end.

We appreciate these considerations and will address them in more detail below.

Major comments

1. The '3D' simulation is used in the BL approximation, BLA (the 1D MYNN simulation has the BLA as an intrinsic restriction). With this, it does not take into account what is considered by some authors (e.g.,Zhong and Chow, 2013, Muñoz-Esparza et al., 2015, Goger et al., 2018) to be the most relevant missing process in BLA schemes in complex terrain, i.e. TKE production due to horizontal shear. Indeed, the 3D PBL (BLA) scheme accounts for horizontal mixing (as the authors claim), but if the (horizontal fraction of) TKE is not adequately produced, the effect of this mixing must be minimal – or even detrimental. I suspect that the almost identical results for the MYNN and 3D PBL (BLA) schemes is to a substantial fraction due to this BLA choice. I think the paper would largely gain, if at least a 'sample day' (as some sort of case study) would be presented and discussed (could be in an appendix or supplemental material).

   Thank you for the suggestion to highlight a sample day. This is something we considered in the original submission. Now, we have added a new Appendix A presenting results from 21 July 2019. The following text from the paper (lines 300-303) explains our choice:

"To expand upon the composite-average wind speed analysis in Figure 4, results from a sample day during the study period, 21 July 2019, are presented in Appendix A. This day was chosen to highlight differences between the 3D PBL and MYNN configurations while also showing consistency with the composite-average results. The same day was highlighted in the original HilFlowS study (Wharton et al., 2015, see Figure 5 therein)."

The Appendix is referenced in the main text when the corresponding figures (4 and 8) are discussed.

We have also expanded our introduction of the BLA (now referred to as the "PBL approximation" for consistency with Juliano et al., 2022), including the suggested references, on lines 171-181:

"Note that following Rybchuk et al. (2022), Arthur et al. (2022), and Wiersema et al. (2023), the PBL approximation (Mellor, 1973; Mellor and Yamada, 1982) is used within the 3D PBL scheme (pbl3d_opt=1) to improve computational efficiency and numerical stability (see discussions therein, and in Juliano et al., 2022). Indeed, the full 3D PBL scheme was found to be computationally unstable in the present domain, likely due to the turbulence length-scale calculation. This was also the case in the complex-terrain studies of Arthur et al. (2022) and Wiersema et al. (2023). With the PBL approximation, the divergences of horizontal turbulence shear stresses and turbulent fluxes are retained in the prognostic equations for momentum and scalars, respectively. However, horizontal gradients are neglected in the system of equations used to calculate the stresses and fluxes, allowing them to be determined analytically. Horizontal gradients are also neglected in the prognostic equation for TKE. Thus, TKE production due to horizontal shear, which has been found by previous studies to be important in complex terrain (Zhong and Chow, 2012; Muñoz-Esparza et al., 2016; Goger et al., 2018), is not considered here. Potential ramifications of using the PBL approximation in this study are discussed further below."

While we acknowledge that it would have been ideal to use the full 3D PBL scheme in this study, it was found to be computationally unstable (as in previous similar studies), and we now state that explicitly.

2. Wind data. I am not familiar with the lidar type used in this study (ZephIR300) – but I trust that the authors use the instruments according to its specifications – with an amazingly high accuracy for a very short averaging time (15 s), and high vertical resolution at the same time. It is mentioned (l. 116) that in an earlier study (Wharton et al., 2015) data was 'corrected' according to some 'Dynamics software' provided by the manufacturer. It is not stated, however, whether this correction was also applied in the present study. Is it? Also, the magnitude of the correction factors are used to estimate 'uncertainty' of the data. I am not sure whether this is a valid approach. Corrections are usually applied to measured data in order to correct for a known deficiency or violation of an assumption. If the correction is well based (and documented), the data is better (more reliable) after correction – irrespective of the (relative) magnitude of the correction. However, in a model evaluation study it must

strictly be distinguished between model errors (what is investigated) and observational errors. If the data is not accurate enough, it cannot be used for model verification (or evaluation).

We have modified the discussion of lidar errors and the Dynamics software from Wharton et al. (2015) (see lines 123-134), noting explicitly that we do not recalculate the conversion factors in the present study:

"An earlier experiment in the APWRA (Wharton et al., 2015) that used identical ZephIR300 lidars to measure hill speedup flows and their effects on power production assessed terrain-induced measurement errors with the Dynamics software package provided by ZephIR Ltd. As discussed therein, the software converts raw lidar line-of-sight velocity data into unbiased measurements of wind speed and wind direction for hilly sites, based on the work of Bingöl et al. (2009). In Wharton et al. (2015), conversion factors for all wind directions and measurement heights ranged from +1% to +8% for the hill lidar, within the range of the Bingöl et al. (2009) study. Moreover, the correction factors associated with the predominant wind direction were closer to zero: +3% for the hill lidar and -2% for the base lidar near the bottom of the hill.

The conversion factors in Wharton et al. (2015) were calculated for a hill that is similar to those at the HilFlowS site, and are presented here for additional context. However, conversion factors are not recalculated for the present study. Rather, the potential +/-10% calculated by Bingöl et al. (2009) is used to conservatively bound the potential mean error in the measured horizontal wind speed. It should be noted that prior to the HilFlowS experiment, the lidars were cross-compared with high agreement (see Wharton and Foster, 2022) providing confidence in their use for model evaluation."

Ultimately, the text above aims to acknowledge potential measurement error while also providing confidence in its use for model evaluation.

3. Observed TKE: if I understand correctly (l.297) the authors determine the velocity variances from only 8 'instantaneous' velocity estimates (every 15 s) over a 2-min period. This of course corresponds to only a small fraction of the total power spectrum and likely means that actual magnitude of TKE is (largely?) underestimated. Possibly, one of the cited observational studies has tested these TKE estimates against true turbulence observations (e.g., from a sonic anemometer)? In any case, data from a sonic anemometer (not necessarily at the same site) could be used to 'model' the chosen approach (i.e., sampling a wind component every 15 s, and calculating the variance according to the chosen approach) and comparing it to the 'full TKE'.

We agree that we took some liberty with our analysis of TKE in the submitted manuscript. The ZX300 gives only a proxy-TKE estimate. It is limited by both the temporal and spatial resolution of the lidar in that it misses the highest frequency turbulence structures. It is not a 3D sonic quality TKE measurement. We would be comfortable calling it "quasi-TKE" if that is preferable to the referee. For now, we refer to the observed TKE as an estimate.

Additionally, while we appreciate the suggestion to "model" the full lidar TKE using sonic data, we believe this is out of scope for the current study. Instead, we have decided to make only qualitative comparisons between the observed and modeled TKE values, and have removed TKE bias calculations from Table 2 and Figure 6. We also focus more on model-to-model TKE comparisons as suggested in the next comment. The modified text on lines 336-343 describes this decision:

"Note that both the observed and modeled TKE values have inherent limitations. The lidar TKE estimates are spatially averaged over the lidar's conical scanning volume and are time-averaged in 10-min windows. Furthermore, the estimated TKE is limited by the 15-s sampling frequency (see additional discussion in Sathe et al., 2011). Lidar TKE estimates are also influenced by complex terrain, as discussed above for wind speeds. The modeled TKE is fully parameterized (i.e., it is assumed that there is no resolved TKE) in each model grid cell and is output as an instantaneous value every 10 min. Ultimately, these limitations preclude direct comparison of observed and modeled TKE values (i.e., bias calculations). In what follows, the time-height structure of the TKE is compared qualitatively between the observations and the model, while only the modeled TKE values are compared quantitatively."

4. The overall statistics (Tab. 2) for TKE are not overly informative (see general considerations). But it is interesting to compare Fig. 6b and 6d. For a given time and height, the 3D PBL TKE scheme produces less TKE than the 1D MYNN scheme (difficult to judge, though, from the colour bar for heights>50 m and nighttime conditions). During the night, both parameterizations underpredict TKE, while during the day the MYNN scheme overpredicts and the 3D scheme still (dominantly) underpredicts. This is at odds with previous experience with 1D turbulence schemes in complex terrain – where usually underprediction is claimed due to neglecting horizontal shear production. As both schemes are employing the BLA (and the TKE observations are not particularly trustworthy, see major comment 3), it is more the relative performance of the two schemes that is interesting. Apparently, the additional (horizontal) mixing in the 3D scheme – and at the same time neglection of the relevant production terms in the TKE equation (BLA) - has an overall detrimental effect on the TKE levels. In this context it is interesting to note that in the original publication of the 3D PBL scheme (Kosović et al., 2020), TKE (i.e., the three velocity variances) were largely underestimated during the day in a complex terrain verification study (their Fig. 5). In the BLA, additional mixing in the 3D PBL parameterization may lead to an unwanted overcompensation. I think this should at least be discussed.

As noted in response to the previous comment, we agree with the referee and have removed TKE bias calculations from Table 2 and Figure 6. We focus instead on the temporal trends of the modeled vs observed TKE, as well as model-to-model comparisons (see lines 344-353):

"In the midday, observed TKE is elevated throughout the lidar's vertical range due to surface heating and associated atmospheric instability. The speedup flows are also accelerating during this time, leading to peak TKE values below 50 m AGL due to shear associated with the jet-like velocity profile (Figure 6a, 1200-1800 PST). Both model configurations capture

elevated TKE during this time (Figure 6b,c). However, the MYNN configuration generally predicts larger TKE values relative to the 3D PBL configuration. This is likely because the 3D PBL scheme with the PBL approximation introduces additional horizontal mixing, relative to MYNN, without added TKE production due to horizontal shear. Reduced TKE in the 3D PBL configuration is associated with improved velocity profile predictions in the midday (see Figure 4, 1200-1500 PST), although the near-surface jet-like flow is not captured accurately by the model. During and after the peak of the speedup flow (1800-0900 PST), the observations and both model configurations show increased TKE near the surface, with reduced values aloft."

We also thank you for pointing out the likely mechanism causing reduced TKE in the 3D PBL configuration, which we have included in the above.

5. Comparison between the MYNN and the 3D-PBL(BLA) schemes. I think it is fair to state that there is no statistically significant difference between the two schemes – at least not when taking the statics as presented into account. If indeed the advantages of the 3D PBL scheme in complex terrain should be evaluated, the statistical information should definitively be extended – and it would probably be advisable to use the full (i.e., non-BLA) 3D PBL scheme.

As discussed in response to RC1 General comments #4 and #5 above, we generally agree that the composite average differences between PBL configurations are small in the present case study. Throughout the paper, we have tried not to overstate the performance of the 3D PBL scheme, but also to highlight the differences we see. We hope that the addition of the sample day in Appendix A helps in this regard.

Lastly, as in response to #1 above, we acknowledge that it would have been ideal to use the full 3D PBL scheme in this study. Although we could not get it to work in this case due to numerical instability, we hope that our added discussion of the BLA and suggestions for future studies (see lines 386-491) are beneficial to the community.

Detailed comments

l.25 '…*referred to more generally as numerical weather prediction (NWP) models':* I don't think this can be said (global NWP models with a grid spacing of some 10 km – and more -  will not qualify as 'meso-scale model'. I suggest to simply delete this part of the sentence….

Please see our response to the comment below…

l.26 'Historically, NWP models….': again, this is a little short history. Historically (in the fifties of the last century – to give history a date), NWP models have started with several hundreds of km as grid-spacing. The present reviewer remembers  the introduction of first so-called 'limited area models' (downscaled from the global models – but only on a limited area) with a grid-spacing of some 20-30 km (and this was thought to be revolutionizing at the time…). In this sentence, only 'or larger' is approximately correct…..

In response to this and the previous comment, we have generalized the discussion of mesoscale/NWP models as follows (lines 25-31):

"…complex terrain is usually under-resolved in mesoscale models, a subset of numerical weather prediction (NWP) models. Historically, NWP models were run with horizontal grid spacing on the order of 10-100 km. However, with ongoing advances in computing power, operational NWP models may now be run at higher resolution. For example, the High-Resolution Rapid Refresh model (HRRR; Benjamin et al., 2016; Dowell et al., 2022), maintained by the National Oceanographic and Atmospheric Administration (NOAA), covers the continental United States with 3 km horizontal grid spacing. Recently, NWP models have been tested with 1 km or sub-kilometer grids (e.g., Olson et al., 2019), but their ability to capture local terrain-driven flow variability at the grid scale or smaller is inherently limited."

Tab 1 'Mfr-$P_R$' (in the title row) is not explained. Similarly, the two variables 'H' and 'D' have not been introduced (even they might be guessed from the context). Finally, 'NREL-2.3' / 'Bonus' etc. need to be explained.

RC1 also noted this in their Specific comment #16 above. We have updated the caption to explain the abbreviations/symbols used in the table. We also refer readers to the text, where we describe/cite the best-available public datasets used for the modeled turbines.

l. 215 (eq. 1): I don't think this eq. defines what usually is called a bias. This is just a difference between a modelled and an observed value of variable 'VAR' at some time and location. Upon averaging over time and/or location (height) this may eventually lead to a bias estimate, i.e. a systematic model deficiency.

In response to RC1 General comment #1 above, we have corrected this and updated the definitions of the error metrics in equations 1-5.

l. 226 '..error metrics are presented in Tab.2': First of all, the caption of Tab 2 must clearly specify that *average bias* is shown. Not in the sense of the previous comment (because bias is always associated with an average (systematic) behavior).Rather, it must become clear that this is a temporal *and* spatial average (this is at least what I must assume when I compare Fig. 3b to the first row of Tab.2). For the spatial (i.e., vertical) average it is essential over which height range the spatial averaging is applied (and why). Having said that, the resulting numbers (close to zero through heavy averaging) are quite useless – and might [heavily] change if the height range over which averaging is applied – or the time - were changed. Also, the error metrics must be explained in the caption or at least a reference must be given where their definition can be found).

Thank you for this suggestion. In Table 2, we now present error metrics averaged over two separate vertical layers, the surface layer and the rotor layer. We believe this helps to distinguish model performance in the near-surface jet-like layer from that in the rotor layer above. Note that the metrics are still time averaged over the full study period. We have also expanded the table caption to explain how the metrics are calculated, and included the same explanation in the text on lines 281-285.

l.238 '…has *only several* model levels….': this is rather unspecific (i.e., more than one but less than 'many'?) – and thus not very helpful.

We have corrected this to be more specific (lines 262-263): "…has only 1-2 model levels ($\Delta z \approx$ 16 m) within the observed jet-like flow layer below roughly 30 m AGL."

Fig. 4, caption: it is stated that 'The shaded regions show ± 1 standard deviation, as well as potential ± 10% error in the observations following'. How is this information combined? The 10% added to the standard deviation? The larger of the two? Another approach? Can the authors be more specific?

We apologize for this confusion. In response to RC1 General comment #2 above, we have better distinguished the potential (mean) lidar error of +/-10% from the composite standard deviation. These two quantities are not combined and are now considered separately in both the figure and the text. Note that in addition to Figure 4, Figures 8, 9, A1, and A2 have been similarly modified.

l.273ff: Following the presentation of results, the authors emphasize the errors in the observations (which is, of course, a little 'bad style' in a model evaluation study: to attribute an important source for the differences to the errors of the observations). It is clear that the authors cannot be made responsible for the observational errors (or uncertainties) -  but when having uncertain data to compare with, the analysis procedure should take this into account (and there are various approaches in the literature how to do this). If the data quality is not good enough, then the data cannot be used for model evaluation.

As discussed above for Major comment #2, we believe the lidar data quality is sufficient for the model evaluation in our study. However, for context, we have tried to acknowledge the potential for measurement error in complex terrain near the beginning of the paper. Ultimately, we do not expect the lidar errors to affect the main conclusions about model evaluation, which we now state more clearly on lines 318-319:

"In general, the expected maximum lidar error is smaller than the model bias, especially near the surface. Thus, the potential lidar error is not expected to affect the present conclusions related to model evaluation."

l.297 calculation of velocity variances: if I understand correctly, there are only 8 values going into the estimation of the variance – and this in a frequency range that only covers a small range in the power spectrum. If the authors would use full-resolution turbulence data (from a sonic anemometer, say) it could be tested (in some sort of model propagator) how large the variance loss actually is under different conditions (will be much smaller under stable conditions than during the day).

As discussed in our response to Major comment #3 above, we believe such an extension of the observed TKE values, while interesting, would be outside the scope of this work. Instead we have focused more on qualitative comparisons between the observed and modeled TKE.

However, we now note explicitly in the text that the estimated lidar TKE is limited by the sampling frequency (see lines 337-338).

l.363    '…reducing the negative bias by as much as 50%'. Looking at Fig. 8 or 9 it is probably fair to add  that after 12 PST it can also [more than] double it.

We have noted this (see line 402).

l. 458 '….larger horizontal gradients will be resolved…..':

Perhaps this comment is incomplete…

l. 610 please correct the reference….

Fixed.

References

Goger B, Rotach MW, Gohm A, Fuhrer O, Stiperski I, Holtslag AAM: 2018, The Impact of 3D Effects on the Simulation of Turbulence Kinetic Energy Structure in a Major Alpine Valley, *Boundary-Layer Meteorol,* **168** (1), 1-27.

Juliano TW, Kosović B, Jiménez P A, Eghdami M, Haupt SE, and Martilli A: 2022, "Gray zone" simulations using a three-dimensional planetary boundary layer parameterization in the Weather Research and Forecasting model, Mon. Wea. Rev., 150, 1585–1619

Kosović B, PA Jiménez, TW Juliano, A Martilli, M Eghdami, A P Barros, and S E Haupt, 2020: Three-dimensional planetary boundary layer parameterization for high-resolution mesoscale simulations. J. Phys.: Conf. Ser., 1452, 012080, https://doi.org/10.1088/1742-6596/1452/1/012080.

Muñoz-Esparza D, Sauer JA, Linn RR, Kosović B (2015) Limitations of one-dimensional mesoscale PBL parameterizations in reproducing mountain-wave flows. J Atmos Sci 73(7):2603–2614

Zhong S, Chow FK: 2013 Meso- and fine-scale modeling over complex terrain: parameterizations and applications. In: Chow FK, De Wekker SFJ, Snyder BJ (eds) Mountain weather research and forecasting, Springer atmospheric sciences. Springer, Berlin, pp 591–653

---

## Referee Report (RR1)

Arthur et al
**Evaluating mesoscale model predictions of diurnal speedup events in the Altamont Pass Wind Resource Area of California**
Manuscript wes-2024-137, first revision

Introduction and recommendation

This manuscript compares simulations of atmospheric flow at a wind power production site in central California, performed with the WRF model at 1-km horizontal grid spacing and two alternative parameterizations for planetary boundary layer (PBL) turbulence. My first review of the manuscript highlighted some deficiencies in the interpretation of forecast verification statistics (systematic vs. random errors). I also recommended de-emphasizing the discussion of the differences between the two PBL schemes, which appeared to be rather marginal in practice. I am generally happy with the revisions, with the single exception of the answer to my minor comment 10, which I do not find entirely satisfactory. I appreciated the addition of Appendix A and Figures A1 and A2, which show the impact of the PBL schemes more convincingly.

I recommend acceptance, provided that lines 260-265 and 290-293 of the revised manuscript are edited as described below.

Specific (minor) comments

1.  One of my comments on the first version of the manuscript (number 10) was about the negative wind speed bias at z < 30 m AGL, visible in Figure 3b. At lines 260-265 of the revised manuscript, the authors interpret the negative bias during speedup events (18-21 LST) using mass continuity arguments:

    *"Conversely, wind speeds are underestimated near the surface, indicating that the model fails to capture near-surface accelerations. ... While the model captures some negative vertical velocities at the study site during the speedup events (see contours in Figure 3f), its vertical velocities are too weak and thus do not translate to near-surface accelerations of the magnitude seen in the observations."*

    This argument might explain why the negative bias becomes slightly larger during the evening speedup events, but not why it persists throughout the whole diurnal cycle. A remark was added in response to my comment, at lines 290-293 of the revised manuscript:

    *"This may be due to slightly improved predictions of vertical mixing of higher momentum from aloft; during this time, prior to jet development, the winds follow a standard quasi-logarithmic profile. The 3D PBL scheme has been shown previously, in idealized tests, to improve model performance during daytime convective conditions (Juliano et al., 2022)"*.

    This explanation misses the point, because it refers to the small difference in bias between the two simulations with different PBL schemes; not to the bias itself, which remains quite large in both cases. A better explanation of the persistent negative wind speed bias might be that "predictions of vertical mixing of higher momentum from aloft" are quite bad even with the 3D-PBL scheme (Figure 3b), which however does a marginally better job than the competitor (Figure 4).

    The issue is not particularly relevant in the wind energy context, because the near-surface layers are essentially irrelevant for wind power harvesting. It might be important in other contexts, though.

---

## Author Response (AR2)

We thank the referees for their careful reading of the revised manuscript and for their recommendations. Please find our responses below in blue. Note that line numbers refer to the main pdf, not the version with changes tracked.

**Response to Referee 1**

**Introduction and recommendation**

This manuscript compares simulations of atmospheric flow at a wind power production site in central California, performed with the WRF model at 1-km horizontal grid spacing and two alternative parameterizations for planetary boundary layer (PBL) turbulence. My first review of the manuscript highlighted some deficiencies in the interpretation of forecast verification statistics (systematic vs. random errors). I also recommended de-emphasizing the discussion of the differences between the two PBL schemes, which appeared to be rather marginal in practice. I am generally happy with the revisions, with the single exception of the answer to my minor comment 10, which I do not find entirely satisfactory. I appreciated the addition of Appendix A and Figures A1 and A2, which show the impact of the PBL schemes more convincingly. I recommend acceptance, provided that lines 260-265 and 290-293 of the revised manuscript are edited as described below.

Thank you, we appreciate your suggestions on both the original manuscript and the revision.

**Specific (minor) comments**

1. One of my comments on the first version of the manuscript (number 10) was about the negative wind speed bias at z < 30 m AGL, visible in Figure 3b. At lines 260-265 of the revised manuscript, the authors interpret the negative bias during speedup events (18-21 LST) using mass continuity arguments:

"Conversely, wind speeds are underestimated near the surface, indicating that the model fails to capture near-surface accelerations. ... While the model captures some negative vertical velocities at the study site during the speedup events (see contours in Figure 3f), its vertical velocities are too weak and thus do not translate to near-surface accelerations of the magnitude seen in the observations."

This argument might explain why the negative bias becomes slightly larger during the evening speedup events, but not why it persists throughout the whole diurnal cycle. A remark was added in response to my comment, at lines 290-293 of the revised manuscript:

"This may be due to slightly improved predictions of vertical mixing of higher momentum from aloft; during this time, prior to jet development, the winds follow a standard quasi-logarithmic profile. The 3D PBL scheme has been shown previously, in idealized tests, to improve model performance during daytime convective conditions (Juliano et al., 2022)".

This explanation misses the point, because it refers to the small difference in bias between the two simulations with different PBL schemes; not to the bias itself, which remains quite large in both cases. A better explanation of the persistent negative wind speed bias might be that "predictions of vertical mixing of higher momentum from aloft" are quite bad even with the 3D-PBL scheme (Figure 3b), which however does a marginally better job than the competitor (Figure 4).

Thank you for clarifying this point. We agree that inadequate vertical mixing is an important component of the model performance in both PBL configurations. To address your comment, we have decided to keep the above lines as is, but to work the suggested point into the existing higher-level discussion on lines 307-326. In particular, lines 307-313 now read as follows:

"Taken together, wind speed error metrics (Table 2), composite-average profiles (Figures 3 and 4), and results from the sample day (Figure A1) suggest that for both model configurations, the predicted amount of vertical mixing is inadequate to transport higher momentum downward from aloft. This results in a persistent negative wind speed bias below roughly 30 m AGL throughout the day. During speedup events, too much momentum remains within the rotor layer. Although both model configurations produce a pronounced jet below hub-height and reduced wind speeds above (Figure 4, 2100-0600 PST), wind speeds are generally overestimated in the rotor layer and underestimated near the surface."

The issue is not particularly relevant in the wind energy context, because the near-surface layers are essentially irrelevant for wind power harvesting. It might be important in other contexts, though.

We hope that our study is of interest to a general atmospheric modeling audience, so we appreciate this consideration.

**Response to Referee 2**

**General considerations**

The authors have adequately addressed the issues raised by the reviewers in the first round. In particular, the error metrics are now correctly defined and the claimed improvement due to the 3D PBL scheme is more carefully (and hence more appropriately) formulated. I was pleased to see the introduction of the case study day (in the appendix) and I think that also the introduction of at least two layers for the determination of error statistics has produced some added value. However, going through the manuscript once more (or for the first time, for the appendix), I detected some issues that still need to be addressed, before the paper can be published. For formal reasons, I have labelled the one concerning the new appendix as 'major' – but as it is straight forward to be addressed, I don't think that it requires an additional review round.

Thank you, we appreciate your suggestions on both the original manuscript and the revision.

**Major comment**

1) New appendix A: this is in fact a very interesting (and also uncovering) additional part of the manuscript. I think it clearly demonstrates the consequences of heavy averaging (in the vertical),

e.g. leading to the numbers in Tab. 2 - even if it is now distinguished between two 'layers'. In particular, it seems to demonstrate that both parameterizations have too much vertical mixing, producing a deep vertical maximum of some 50 m depths, while the observations show this 'speedup layer' to be about half as deep. While I think that the overall 'conclusion' ('..highlighting potential model improvements...') is rather optimistic, I only comment on the obvious error: in Fig. A1 the lowest observation is at zero m AGL, thus suggesting that at times there is a (mean hourly) wind speed of >6 m/s at the ground. This must be corrected to 10 m (first range gate of the lidar). This also affects Fig. 4 (again, I have to say that I overlooked this in the first review), and possibly Fig. 3a, c.

Thank you for catching this subtle point that was not fully explained in the manuscript. Figures 3, 4, and A1 include observed wind speed and direction data from the lidar's on-board meteorological station at 1 m AGL. These data are shown to provide additional context for the shape of the near-surface jet. However, model errors are not evaluated at this height because extrapolation below the lowest model grid point (i.e., the lowest half level at which velocities are calculated) would be required. We now explain this on lines 240-242 when introducing the error metrics:

"Although several figures herein present observed wind speed and direction from the lidar's onboard meteorological station at 1 m AGL, model errors are not evaluated at this height because extrapolation below the first half level (at roughly 8 m AGL) would be required."

An abbreviated note has also been added to the three figure captions.

**Detailed comments**

Fig. 3d I overlooked in the first review that wind direction bias (panel d) is given in m/s instead of  $^{\circ}$ .

**This has been corrected.**

Tab. 2, caption (and l. 283): why is the 30 m level included in both averaging ranges? As it corresponds to the given definition of the 'rotor layer', it should probably be removed from what is called the 'surface layer', which does not correspond to the traditional surface layer – lowest 10% of the BL height – but possibly to the Inner Layer in the linearized theory of Hunt and colleagues (see the summary paper of Belcher and Hunt (1998) for the references).

We have corrected the vertical averaging ranges for Table 2 such that the 30 m level is only included in the rotor layer. As seen in the diffed pdf, this correction results in minor changes to some of the near-surface metrics, which tend to accentuate the differences between the two layers. We have also generalized the wording to prevent confusion with the layers defined by boundary-layer theory, in both the Table 2 caption and in the text on lines 283-286:

"The metrics shown in the table are time averaged over the full study period and vertically averaged over two separate layers: the rotor layer (lidar measurement heights of 30, 38, 50, 60,

70, 80, 90, and 120 m AGL) and a near-surface layer below the rotor layer (lidar measurement heights of 10 and 20 m AGL)."

**References**

Belcher S.E., Hunt J.C.R. (1998) TURBULENT FLOW OVER HILLS AND WAVES. Annual Review of Fluid Mechanics 30, 507-538, doi: 10.1146/annurev.fluid.30.1.507.